# Exploring Data-Free LoRA Transferability for Video Diffusion Models

Yuchen Wang [1]  Wenliang Zhong [1]  Lichen Bai [1]  Zikai Zhou [1]  Shitong Shao [1]
Bojun Cheng [1]  Shuo Chen [2]  Shuo Yang [3]  Zeke Xie [1]

## Abstract

Video diffusion models leveraging step distillation or causal distillation have achieved remarkable performance. However, adapting existing LoRAs to these variants remains a critical challenge due to weight space mismatches. We observe that direct application leads to style degradation and structural collapse, yet the underlying mechanisms remain poorly understood. To fill this gap, we delve into the weight space and identify that the incompatibility stems from spectral interference within shared functional clusters defined over singular subspaces. Specifically, our analysis reveals that while both paradigms respect spectral rigidity, they establish conflicting routing pathways that clash through constructive overload or destructive cancellation. To address this issue, we propose Cluster-Aware Spectral Arbitration (CASA), a data-free framework that dynamically arbitrates between safeguarding the target's manifold and restoring LoRA alignment based on spectral density. Extensive experiments demonstrate that CASA effectively mitigates artifacts and revives LoRA functionality. Our code is available at https://github.com/Noahwangyuchen/CASA.

## 1. Introduction

Video diffusion models (VDMs; Wan et al., 2025; Kong et al., 2024; Brooks et al., 2024; Jiang et al., 2025) have recently emerged as a powerful paradigm for high-fidelity video generation, enabling coherent synthesis across both spatial and temporal dimensions. However, this capability comes with substantial inference cost, as these models typically requires iterative denoising while jointly processing tokens from all video frames. To reduce this cost, a

[1]The Hong Kong University of Science and Technology (Guangzhou) [2]Nanjing University [3]Harbin Institute of Technology (Shenzhen). Correspondence to: Zeke Xie <zekexie@hkust-gz.edu.cn>.

*Proceedings of the $43^{rd}$ International Conference on Machine Learning*, Seoul, South Korea. PMLR 306, 2026. Copyright 2026 by the author(s).

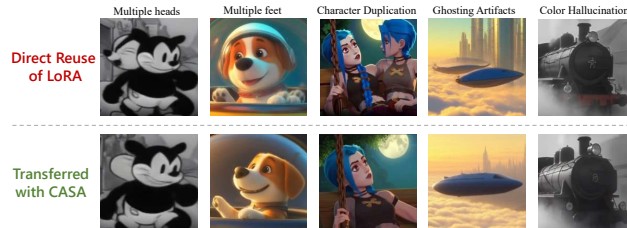

*Figure 1.* Failure modes under direct LoRA reuse on distilled VDMs (top) and the results after CASA transfer (bottom).

series of efficient variants have been proposed (Shao et al., 2026), including step distillation (Zhang et al., 2025; Ding et al., 2025; Lin et al., 2025) and various causal distillation strategies (Huang et al., 2025; Gao et al., 2025; Yin et al., 2025). These approaches are typically implemented via full fine-tuning (FFT) of a pretrained base model, resulting in distilled VDMs that preserve generation quality while significantly accelerating inference. As a consequence, modern VDM ecosystems increasingly consist of families of closely related models that share a common generative foundation but differ in their weight space due to fine-tuning.

In parallel, Low-Rank Adaptation (LoRA) has become a standard tool for efficient and controllable adaptation of large generative models (Hu et al., 2022). However, when LoRAs trained on a base video diffusion model are applied directly to its distilled variants, they are prone to degrading or generating various artifacts, as shown in Figure 1. Retraining LoRAs on each distilled model is computationally expensive and requires access to user data, rendering it impractical in many real-world scenarios.

Resolving this incompatibility requires a mechanistic understanding of how FFT and LoRA modify VDMs. However, the effects of FFT and LoRA in VDMs have not been systematically studied. In this work, we fill this gap by conducting the first weight-space analysis of fine-tuning in video diffusion models. Our analysis reveals a consistent spectral rigidity property across layers and further shows that adaptation primarily manifests as structured rotations of singular subspaces that exhibit strong cluster-level coherence.

Building on these insights, we further utilize a routing-based perspective for characterizing the effect of FFT and LoRA, showing that the failure of LoRA reuse on distilled video

diffusion models stems from spectral interference between incompatible routing patterns. While both adaptations respect spectral rigidity, they establish conflicting functional routes in the shared weight space, leading to style degradation and structural artifacts. To resolve this conflict, we propose Cluster-Aware Spectral Arbitration (CASA), a data-free framework that formulates LoRA transfer as a cluster-level arbitration problem in the spectral domain. CASA is designed to dynamically balance the preservation of generative pathways of the target model with the restoration of LoRA functionality. As a result, CASA enables effective LoRA reuse in distilled VDMs without additional training or access to user data. The contributions of this work can be summarized as follows:

- To the best of our knowledge, we provide the first weight-space analysis of full fine-tuning and LoRA in VDMs, and introduce a routing-based perspective for characterizing their effects in singular subspaces.
- We identify the root cause of LoRA failure on distilled video diffusion models as spectral interference between incompatible routing patterns introduced by full fine-tuning and LoRA.
- We propose Cluster-Aware Spectral Arbitration (CASA), a data-free method that enables effective reuse of LoRAs on distilled models without additional training or user data.

## 2. Related Works

**Video Diffusion Models and Distillation.** Video diffusion models (VDMs; Wan et al., 2025; Blattmann et al., 2023; Yang et al., 2025b; Rombach et al., 2022) have recently demonstrated strong performance in high-fidelity video generation. However, these models largely rely on bidirectional attention processing tokens from all video frames simultaneously, incurring high computational cost and preventing streaming generation. To alleviate this, step distillation methods (Ding et al., 2025; Lin et al., 2025; Xi et al., 2025) aim to reduce the number of denoising steps by training a student model to approximate the multi-step behavior of a pretrained teacher. In parallel, causal distillation methods (Huang et al., 2025; Lu et al., 2025; Yang et al., 2025a; Gu et al., 2025) reformulate bidirectional video diffusion into causal autoregressive processes, enabling streaming generation with significantly reduced latency. Despite their algorithmic differences, these approaches are typically implemented via full fine-tuning of a pretrained base model (Huang et al., 2025; Zhang et al., 2025), resulting in distilled variants that preserve generation quality while altering the underlying weight space. In this work, we analyze how such full fine-tuning reshapes the weight space of video diffusion models and how it interacts with parameter-efficient adaptations such as LoRA.

**Weight Space Analysis of Model Adaptation.** Understanding how fine-tuning modifies pretrained models in the weight space has attracted increasing attention. Prior works (Liu et al., 2024; Fan et al., 2025; Meng et al., 2024; Si et al., 2025b; Shuttleworth et al., 2025; Si et al., 2024; 2025a) have analyzed the effect of full fine-tuning and LoRAs from a weight space perspective, often by leveraging singular value decomposition (SVD) to study spectral properties induced by adaptation. However, existing analyses are largely limited to LLMs and do not directly extend to VDMs with fundamentally different generation dynamics. To the best of our knowledge, a systematic weight-space analysis of full fine-tuning and LoRA in video diffusion models remains largely unexplored.

**LoRA Transferability.** Low-Rank Adaptation (LoRA) (Hu et al., 2022) and its variants (Dettmers et al., 2023; Kopiczko et al., 2024; Zhang et al., 2023) have emerged as an efficient and widely adopted parameter-efficient fine-tuning technique. Recent work has explored LoRA transfer across models from different perspectives. X-Adapter (Ran et al., 2024) learns mapping layers between source and target models to align feature spaces for PEFT modules. Trans-LoRA (Wang et al., 2024) relies on synthetic data generated by large language models to facilitate cross-model transfer. Other methods aim to improve intrinsic transferability by constraining the update space, such as LoRA-X (Farhadzadeh et al., 2025a), which restricts updates within selected singular directions. More closely related to our setting, ProLoRA (Farhadzadeh et al., 2025b) enables data-free LoRA transfer by projecting source adaptations into the target weight space. Despite these advances, existing methods are primarily developed for large language models or image generation models. In contrast, we study LoRA transfer in VDMs from a weight-space perspective and propose a data-free transfer method grounded in the specific spectral structure of VDMs.

## 3. Analysis

In this section, we analyze how FFT and LoRA modify VDMs in the weight space and how their interaction leads to LoRA incompatibility on distilled models. We first reveal a consistent spectral rigidity across layers, then show that adaptation mainly manifests as structured, cluster-coherent perturbations of singular subspaces. Finally, we characterize FFT and LoRA routing patterns at the cluster level and identify spectral interference as the key failure mechanism. Together, our analysis establishes a mechanistic foundation for understanding LoRA incompatibility on distilled VDMs. We conduct the analysis using Wan2.1-T2V-1.3B (Wan et al., 2025) as the source model, FastWan2.1-T2V-1.3B (Zhang et al., 2025) as the distilled target model, with Jinx-v2 and Steamboat-Willie-1.3B as LoRAs.

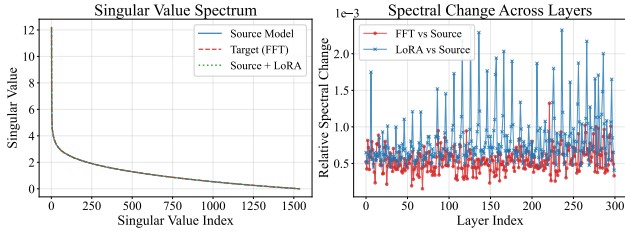

*Figure 2. left:* Singular values of one layer from the source model and its FFT and LoRA counterpart. *right:* Their relative spectral changes across all layers.

## 3.1. Spectral Rigidity

We first examine the global spectral effects of FFT and LoRA on video diffusion models. For each weight matrix, we compare the singular value spectrum of the base model with that of its FFT- and LoRA-adapted counterparts. The left part of Figure 2 shows singular value curves of a random layer, where we observe that the spectral shape is largely preserved under both adaptation strategies. To quantify this observation at scale, we measure the relative spectral change across all layers of the model using the $\ell_2$ norm,

$$\rho_2 = \frac{\|\mathbf{S}' - \mathbf{S}\|_2}{\|\mathbf{S}\|_2}, \tag{1}$$

where $\mathbf{S}$ and $\mathbf{S}'$ denote the singular values before and after fine-tuning, respectively. As presented in the right part of Figure 2, both adaptations exhibit extremely small relative spectral changes not exceeding $0.3\%$, indicating that fine-tuning does not substantially redistribute spectral energy.

These results reveal a pronounced spectral rigidity property in video diffusion models: despite significant differences in training objectives and optimization procedures, both FFT and LoRA preserve the singular value spectrum to a high degree. This suggests that fine-tuning does not alter the overall capacity allocation or destabilize the underlying generative manifold, but instead induces more subtle structural modifications. At the same time, rigidity in the singular values implies that the primary effects of fine-tuning must arise from changes in the associated singular subspaces. In the following subsection, we therefore turn to analyze how FFT and LoRA perturb these subspaces and show that such perturbations exhibit strong structured patterns.

## 3.2. Structured Perturbations of Singular Subspaces

To characterize how fine-tuning alters the internal representation of video diffusion models, we analyze the changes induced by FFT and LoRA at the level of singular subspaces. We quantify subspace alignment using the cosine similarity matrix $|\mathbf{U}^\top \mathbf{U}'|$, where $\mathbf{U}$ and $\mathbf{U}'$ denote the left singular bases before and after fine-tuning, respectively. Unless otherwise stated, we focus on $\mathbf{U}$, while observing qualitatively

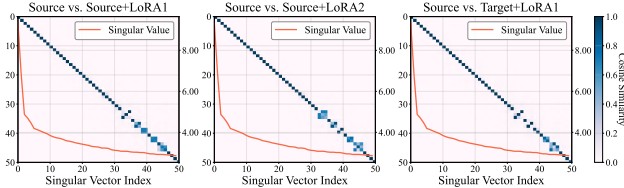

*(a)* Head singular vectors remain highly steady.

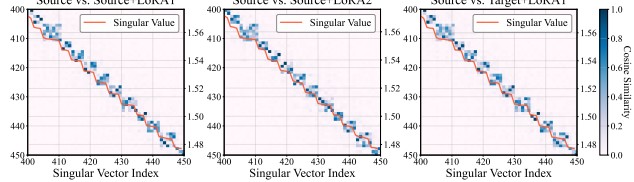

*(b)* Middle spectrum exhibits block-wise mixing aligned with step-like spectral plateaus.

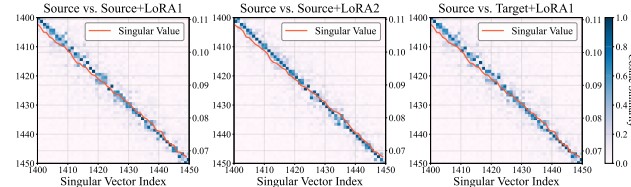

*(c)* Tail shows increasingly diffuse mixing due to near-degeneracy.

*Figure 3.* Similarity matrix of left singular bases before and after fine-tuning. The pattern is consistent across LoRAs and persists after applying LoRA to the distilled model. Please refer to Appendix A for more examples.

identical behavior for $\mathbf{V}$.

Figure 3 visualizes this similarity matrix together with the corresponding singular value spectrum of a random layer. We examine three representative regions along the spectrum, corresponding to head, middle, and tail, to illustrate the structure of subspace perturbations. In the head of the spectrum, where singular values are dominant and well separated, the similarity matrix is sharply concentrated along the diagonal. Each singular direction with top singular values remains closely aligned with its original counterpart, indicating that fine-tuning preserves the leading subspace structure almost identically. This behavior contrasts with observations in large language models, where LoRA has been shown to introduce intruder dimensions that exhibit extremely low cosine similarity to any pre-trained singular direction (Shuttleworth et al., 2025), or to significantly increase the leading singular values (Si et al., 2025b).

Moving to the middle of the spectrum, the similarity matrix exhibits clear block-wise patterns. Instead of a strict one-to-one correspondence, groups of neighboring singular vectors display mutual similarity, forming coherent clusters. Notably, these clusters align with step-like plateaus in the singular value curve, where local spectral gaps separate adjacent groups. Within each plateau, singular directions are more interchangeable under fine-tuning, while mixing

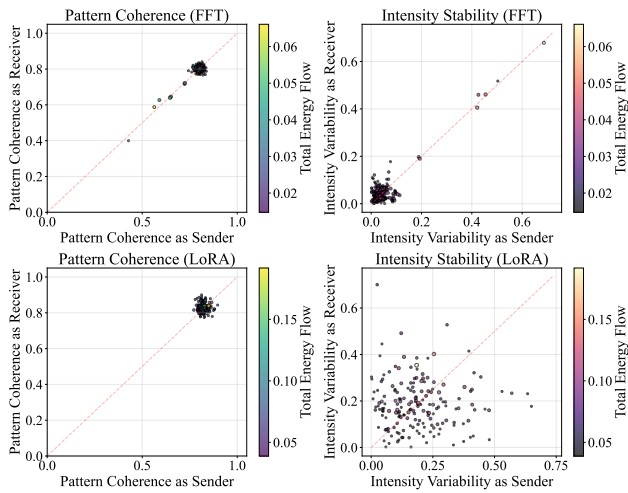

*Figure 4.* Routing pattern and energy coherence of clusters.

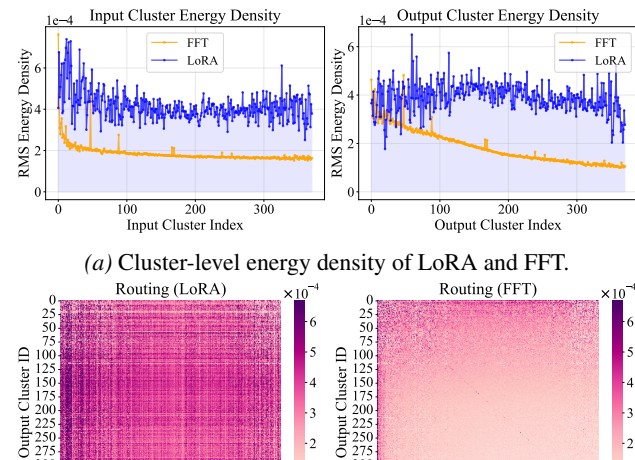

*(a)* Cluster-level energy density of LoRA and FFT.

*(b)* Cluster-level routing heatmap of LoRA and FFT.

*Figure 5.* Typical cluster-level routing behaviors of LoRA and FFT. Please see Appendix B.1 for more results.

across plateaus remains limited. In contrast, the tail of the spectrum shows a different behavior. Here, the singular values decay smoothly with much smaller local gaps, and the similarity matrix transitions into a diffuse, banded pattern. Many singular directions in this region are near-degenerate and therefore admit broader mixing under fine-tuning, resulting in the observed spread of similarity.

Importantly, the resulting structure is highly consistent across distinct LoRAs and when applying LoRA to the distilled model, indicating that it reflects a stable organization of the underlying model rather than an artifact of a specific adaptation. Overall, these observations are consistent with classical perturbation theory, which relates subspace stability to local spectral separation (Davis & Kahan, 1970). Regions with larger spectral gaps exhibit more localized and structured mixing, while near-degenerate regions allow more distributed perturbations.

Together, these results indicate that fine-tuning induces structured perturbations across the singular subspaces, giving rise to stable cluster-level organization in the leading and intermediate spectrum. In the following, we build upon this cluster-level view to analyze how FFT and LoRA establish routing patterns, and how their interaction within these clusters leads to incompatible spectral interference.

### 3.3. Cluster-level Routing

To analyze how FFT and LoRA modify the functional interactions within video diffusion models, we study the changes induced in the weight space through a routing perspective. Given a weight update $\Delta$ applied to a linear projection with singular value decomposition $\mathbf{W} = \mathbf{U}\mathbf{S}\mathbf{V}^\top$, we define the corresponding routing matrix as $\mathbf{C} = \mathbf{U}^\top \Delta \mathbf{V}$. Each entry $\mathbf{C}_{ij}$ quantifies how the update introduces the interaction between the $j$-th right singular direction and the $i$-th left

singular direction, providing a natural view of fine-tuning as cross-subspace routing, where columns of $\mathbf{C}$ correspond to senders in the input singular basis, while rows correspond to receivers in the output basis. The magnitude of each entry reflects the strength of information flow induced between the corresponding subspaces.

While routing can be examined at the level of individual singular directions, a meaningful characterization of global routing behavior requires assessing its consistency within the singular clusters identified above. We first group singular directions into clusters based on their mixing before and after fine-tuning[1]. For each cluster, we measure pattern coherence as the cosine similarity between routing directions of singular vectors within the same cluster, and quantify intensity stability as the coefficient of variation of their routing energy, both when acting as senders and receivers. As shown in Figure 4, both FFT and LoRA exhibit strong cluster-level pattern coherence with highly aligned routing directions. At the same time, routing intensity remains stable within clusters, as reflected by low intra-cluster variability in both sending and receiving energy. These results indicate that both methods maintain strong intra-cluster consistency, confirming that singular clusters serve as stable functional units for routing analysis.

We next examine the routing patterns at the cluster level. For each cluster, we measure the root mean square (RMS)

---

[1]We construct a graph over singular vectors before and after fine-tuning, where two directions are connected if their cosine similarity exceeds a threshold of $0.2$, and obtain connected components as clusters, considering only the top-$k$ singular vectors that cumulatively capture $90\%$ of the spectral energy.

energy density as senders and as receivers, normalized by the number of connections within the cluster. Figure 5a shows a typical cluster-wise energy profile. We observe that under FFT, routing energy is highly concentrated in a small number of clusters, which predominantly correspond to the head of the spectrum, indicating their dominant role in shaping the effective generative subspace[2]. These clusters exhibit significantly higher RMS energy as both senders and receivers, while energy density gradually decays toward later clusters. In contrast, LoRA does not exhibit such behavior. Its routing energy is distributed more uniformly across clusters, with no small subset of clusters dominating either outgoing or incoming signal intensity. This pattern is further reflected in the block-wise energy maps presented in Figure 5b, where LoRA induces broadly distributed connections across cluster pairs, while FFT concentrates energy in a limited set of high-intensity blocks.

Together, these observations indicate that FFT establishes a strongly centralized routing structure anchored at a few dominant clusters, while LoRA injects functional modifications in a more diffuse and globally distributed manner.

### 3.4. Cluster-level Routing Interference

We now analyze how LoRA and FFT interact at the cluster-routing level. We first quantify their routing overlap at the cluster level. For each input–output cluster pair, we compute the product of their RMS routing energy, which measures the intensity of co-activation on the same routing pathway. This quantity reflects the potential for interaction, but does not by itself determine whether the interaction is constructive or destructive. We therefore complement this analysis by examining the directional alignment between LoRA and model drift on the same cluster pairs. Interference arises when strong routing overlap coincides with incompatible directional alignment, indicating conflicting updates on the same functional pathway.

Figure 6 visualizes these two quantities. We observe that strong interactions are highly localized, concentrating on a small subset of cluster pairs. Notably, these high-interaction regions predominantly involve head clusters, consistent with their elevated routing energy under FFT. The corresponding direction map exhibits substantial misalignment. Across interacting cluster pairs, LoRA and FFT can be either strongly aligned or strongly opposed, with no global tendency toward constructive or destructive alignment. Importantly, interference in head clusters is particularly consequential. Since these clusters dominate the effective generative subspace, conflicting routing signals injected by LoRA into pathways already strongly modulated by distillation are more likely to perturb the generation behavior.

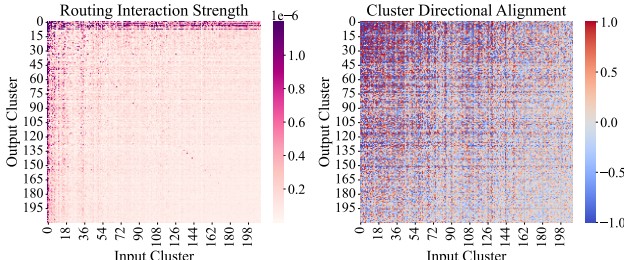

*Figure 6. left:* Cluster-level routing energy overlap strength. *right:* Directional alignment between LoRA and FFT.

Together, these observations suggest that LoRA incompatibility on distilled video diffusion models stems from cluster-level interference. When LoRA injects signals into cluster pathways that are already strongly modulated by model distillation, their interaction can become spectrally incompatible, leading to unstable or distorted generation behavior. This insight suggests that a uniform transfer strategy is doomed to fail; instead, an arbitration mechanism is needed to distinguish between these spectrally conflicting regions.

## 4. Methodology

**Overview and Motivation.** Given a source VDM equipped with a LoRA and its distilled variant obtained by FFT, our goal is to reuse the original LoRA on the distilled model without additional training or user data. Motivated by our analysis, we operate in the singular subspace view, where adaptation is expressed as structured routing, and incompatibility arises from conflicting routing in the same functional subspaces. We therefore propose Cluster-Aware Spectral Arbitration (CASA), a principled routing arbitration framework that decomposes LoRA transfer into two complementary objectives: (i) preserving LoRA-induced functional routing in non-critical spectral regions of the distilled model, and (ii) preventing over-activation along dominant generative pathways shaped by distillation.

### 4.1. Cluster-Aware Routing Representation

Let $\mathbf{W}_s$ denote the weight matrix of a layer in the source model, with singular value decomposition $\mathbf{W}_s = \mathbf{U}_s \mathbf{S}_s \mathbf{V}_s^\top$. Let $\Delta_{\mathrm{lora}} = \mathbf{BA}$ be the LoRA update trained on the source model, and let $\Delta_{\mathrm{fft}}$ denote the weight drift induced by FFT for the corresponding layer.[3]

We project both updates into the singular bases of the source layer and obtain their routing representations by

$$\mathbf{C}_{\mathrm{lora}} = \mathbf{U}_s^\top \Delta_{\mathrm{lora}} \mathbf{V}_s, \quad \mathbf{C}_{\mathrm{fft}} = \mathbf{U}_s^\top \Delta_{\mathrm{fft}} \mathbf{V}_s. \quad (2)$$

Here, each entry $\mathbf{C}(i, j)$ quantifies a routing connection from the $j$-th right singular direction (sender) to the $i$-th left

---

[2]We further explore the generative subspace from the routing perspective in Appendix B.2.

[3]In practice, $\Delta_{\mathrm{fft}}$ can be obtained by subtracting the source weights from the distilled weights at the same layer.

singular direction (receiver).

Following our analysis, we further construct cluster-level routing representations. We first restrict clustering to the leading singular subspace by selecting the smallest $k$ that captures 90% of the spectral energy, i.e.,

$$\frac{\sum_{i=1}^{k} \sigma_i^2}{\sum_i \sigma_i^2} \geq 0.9, \tag{3}$$

where $\sigma_i$ represents the $i$-th singular value. This focuses CASA on the spectrally stable area where structured cluster organization is most pronounced. We then build singular clusters in the top-$k$ subspace using a gap-aware perturbation graph. For indices $i, j \leq k$, we define a predicted rotation strength

$$\mathbf{R}(i, j) = \frac{|\mathbf{C}_{\text{lora}}(i, j)|}{|\sigma_i - \sigma_j| + \epsilon}, \tag{4}$$

and connect $i$ and $j$ if $\mathbf{R}(i, j)$ exceeds a threshold $\tau$. Connected components of the resulting undirected graph define clusters $\{\mathcal{G}_m\}_{m=1}^{M}$. This construction can be interpreted as grouping singular directions whose relative coupling strength exceeds their local spectral separation, consistent with our analysis in Section 3.2.

## 4.2. Cluster-Aware Spectral Arbitration

CASA produces an updated routing matrix $\mathbf{C}_{\text{casa}}$ by combining $\mathbf{C}_{\text{lora}}$ and $\mathbf{C}_{\text{fft}}$ with selective arbitration between reviving LoRA function and protecting the generative subspace of the target model.

**Identifying Spectrally Dominant Routing Constraints.** We characterize spectrally dominant routing regions as implicit constraints imposed by the distilled model, quantified via routing energy density in the singular routing space. Specifically, for each cluster $\mathcal{G}_m$, we compute its sending and receiving density as

$$\rho_m^{\text{send}} = \frac{1}{|\mathcal{G}_m|} \sum_{i \in \mathcal{G}_m} \|\mathbf{C}_{\text{fft}}(:, i)\|_2, \tag{5}$$

$$\rho_m^{\text{recv}} = \frac{1}{|\mathcal{G}_m|} \sum_{i \in \mathcal{G}_m} \|\mathbf{C}_{\text{fft}}(i, :)\|_2. \tag{6}$$

Clusters whose sending or receiving density exceeds a quantile threshold $q_{\text{dom}}$ are regarded as dominant sending or receiving clusters, denoted as $\mathcal{G}_{\text{dom}}^{\text{send}}$ and $\mathcal{G}_{\text{dom}}^{\text{recv}}$, respectively. This criterion captures routing pathways that are heavily utilized by the distilled model and therefore play a central role in its effective generative subspace[4].

---

[4] Please refer to Appendix B.2 for more analysis.

Formally, we define an indicator function $\mathcal{D}(i, j)$ to determine whether a routing entry $(i, j)$ lies in a dominant routing region:

$$\mathcal{D}(i, j) = \mathbb{I}\big[i \in \mathcal{G}_{\text{dom}}^{\text{recv}} \ \vee \ j \in \mathcal{G}_{\text{dom}}^{\text{send}}\big], \tag{7}$$

where $\mathbb{I}[\cdot]$ is the indicator function. A routing entry is considered dominant if $\mathcal{D}(i, j) = 1$.

**Restoring LoRA in Non-Dominant Regions.** For routing entries outside dominant regions, we directly restore the LoRA update by compensating for the FFT drift to avoid destructive cancellation. For entries satisfying $\mathcal{D}(i, j) = 0$:

$$\mathbf{C}_{\text{casa}}(i, j) = \mathbf{C}_{\text{lora}}(i, j) - \mathbf{C}_{\text{fft}}(i, j). \tag{8}$$

Under the assumption that these regions are weakly coupled to the distilled model's generative manifold, this operation corresponds to the minimum-interference solution that exactly restores LoRA-induced routing while leaving dominant pathways unchanged.

**Arbitration in Dominant Routing Regions.** Routing entries that fall within dominant regions, i.e., those satisfying $\mathcal{D}(i, j) = 1$, require more cautious treatment due to their influence on the generative subspace. As analyzed in Section 3.4, directly restoring LoRA updates in these regions may lead to over-activation, disrupting critical generation pathways. Specifically, over-activation can be viewed as a form of constructive interference in routing space, where LoRA and FFT induce aligned updates along the same functional subspace, resulting in disproportionate energy amplification beyond the operating regime of the distilled model.

To capture this behavior, we model the risk of over-activation as a factorized score consisting of a local interaction term and a cluster-level contextual alignment term. We first define the local interaction matrix $\mathbf{E}$ for dominant routing regions as

$$\mathbf{E}(i, j) = \max\big(0, \ \mathbf{C}_{\text{lora}}(i, j) \mathbf{C}_{\text{fft}}(i, j)\big) \cdot \mathcal{D}(i, j), \tag{9}$$

which assigns positive interaction energy only to routing entries where LoRA and full fine-tuning induce aligned routing directions. Entries with opposite directions yield zero interaction energy and are therefore excluded from further consideration.

To incorporate the cluster-level context, we further compute the directional alignment between $\mathbf{C}_{\text{lora}}$ and $\mathbf{C}_{\text{fft}}$. For each pair of clusters $(\mathcal{G}_m, \mathcal{G}_n)$, we define their routing blocks as submatrices

$$\mathbf{C}_{\text{lora}}^{(m,n)} = \mathbf{C}_{\text{lora}}[\mathcal{G}_m, \mathcal{G}_n], \ \mathbf{C}_{\text{fft}}^{(m,n)} = \mathbf{C}_{\text{fft}}[\mathcal{G}_m, \mathcal{G}_n]. \tag{10}$$

The directional alignment between the two routing blocks is then measured by cosine similarity:

$$\text{Cos}(\mathcal{G}_m, \mathcal{G}_n) = \frac{\langle \mathbf{C}_{\text{lora}}^{(m,n)}, \mathbf{C}_{\text{fft}}^{(m,n)} \rangle}{\|\mathbf{C}_{\text{lora}}^{(m,n)}\|_F \|\mathbf{C}_{\text{fft}}^{(m,n)}\|_F + \epsilon}, \tag{11}$$

where $\langle \cdot, \cdot \rangle$ denotes the Frobenius inner product. This cluster-level alignment is then propagated back to individual routing entries as a contextual factor. Let $g(\cdot)$ map a singular index to its cluster id in $\{\mathcal{G}_m\}_{m=1}^M$ within the top-$k$ subspace. We define

$$\text{Context}(i,j) = \begin{cases} \text{Cos}(\mathcal{G}_{g(i)}, \mathcal{G}_{g(j)}), & i \le k, \ j \le k, \\ 1, & \text{otherwise,} \end{cases} \tag{12}$$

and obtain the score of over-activation risk as

$$\mathbf{S}(i,j) = \mathbf{E}(i,j) \cdot \text{Context}(i,j). \tag{13}$$

For entries with $\mathbf{S}(i,j)$ exceeding a quantile threshold $q_{\text{act}}$, CASA applies a magnitude-based arbitration rule that restricts the recovered routing strength to a safe range by retaining only the stronger contribution between LoRA and FFT:

$$\begin{aligned} \mathbf{C}_{\text{casa}}(i,j) = \ &\max\big(|\mathbf{C}_{\text{lora}}(i,j)|, \ |\mathbf{C}_{\text{fft}}(i,j)|\big) \\ &\cdot \text{sign}\big(\mathbf{C}_{\text{lora}}(i,j)\big) - \mathbf{C}_{\text{fft}}(i,j). \end{aligned} \tag{14}$$

All remaining entries in dominant regions are kept as $\mathbf{C}_{\text{lora}}(i,j)$, preserving the FFT-induced generative structure while avoiding widespread suppression of LoRA routing. Finally, we obtain the updated LoRA parameters by projecting $\mathbf{C}_{\text{casa}}$ back to the source weight space as $\Delta_{\text{casa}} = \mathbf{U}_s \mathbf{C}_{\text{casa}} \mathbf{V}_s^\top$, and applying a low-rank factorization to recover $(\mathbf{B}, \mathbf{A})$.

From a unified perspective, CASA can be interpreted as a constrained routing recovery problem in the singular subspace. It constructs an update $\mathbf{C}_{\text{casa}}$ such that the effective routing $\mathbf{C}_{\text{fft}} + \mathbf{C}_{\text{casa}}$ recovers $\mathbf{C}_{\text{lora}}$ as closely as possible, while respecting the stability constraints imposed by the distilled generative subspace. To this end, CASA enforces routing compensation in non-dominant regions, where interactions are weakly coupled to generation, and restricts the recovered routing in spectrally dominant regions to remain within a safe activation envelope. This yields a minimal-intervention solution that restores LoRA functionality while preserving the generative space of the distilled model.

## 5. Experiments

In this section, we will evaluate the effectiveness of CASA on the task of transferring LoRAs trained on a base video diffusion model to its distilled variants. Moreover, we conduct ablation study and analysis to validate the design choices. Additional analyses are provided in Appendix D.

### 5.1. Experimental Settings

**Models.** We conduct experiments on two scales of video diffusion models based on the Wan2.1 (Wan et al., 2025) text-to-video architecture. We use Wan2.1-T2V-1.3B and

*Table 1.* Comparison of direct LoRA reuse and transferring LoRA with CASA on distilled VDMs. Best results are bold.

| LoRA | Target Model | Method | Quality Score | CSD (%) |
|---|---|---|---|---|
| Steamboat-Willie-1.3B | FastWan2.1-T2V-1.3B | Direct Reuse | 1.27 | 78.35 |
| | | CASA | **1.58** | **81.49** |
| | Rolling Forcing | Direct Reuse | 2.31 | 71.03 |
| | | CASA | **2.45** | **71.81** |
| Jinx-v2 | FastWan2.1-T2V-1.3B | Direct Reuse | 1.46 | 68.17 |
| | | CASA | **1.51** | **70.28** |
| | Rolling Forcing | Direct Reuse | **2.69** | 71.25 |
| | | CASA | 2.67 | **73.80** |
| Film-Noir | FastWan2.1-T2V-14B | Direct Reuse | 1.90 | 60.18 |
| | | CASA | **2.03** | **61.52** |
| | Krea Realtime Video | Direct Reuse | 2.92 | 66.08 |
| | | CASA | **2.95** | **67.22** |
| Steamboat-Willie-14B | FastWan2.1-T2V-14B | Direct Reuse | 1.86 | **63.47** |
| | | CASA | **1.95** | 62.56 |
| | Krea Realtime Video | Direct Reuse | **2.04** | 70.06 |
| | | CASA | 2.00 | **72.27** |

Wan2.1-T2V-14B as source models on which LoRAs are trained. For Wan2.1-T2V-1.3B, we evaluate LoRA transfer to two distilled variants, including FastWan2.1-T2V-1.3B (Zhang et al., 2025) and Rolling Forcing (Liu et al., 2025b). For Wan2.1-T2V-14B, we consider FastWan2.1-T2V-14B (Zhang et al., 2025) and Krea Realtime Video (Millon, 2025) as target models. The target models cover both step distillation and causal distillation strategies.

**Datasets.** We evaluate CASA using a diverse set of LoRAs. For experiments on Wan2.1-T2V-1.3B, we use Steamboat-Willie-1.3B and Jinx-v2. For Wan2.1-T2V-14B, we adopt Film-Noir and Steamboat-Willie-14B. All LoRAs are obtained from public Hugging Face repositories. Detailed model card links are provided in Appendix C.1.

**Metrics.** We evaluate both generation stability and LoRA style preservation using complementary metrics. For generation quality, we adopt VideoAlign (Liu et al., 2025a) to obtain the visual quality and motion quality, reporting their average as a Quality Score. To evaluate style transfer fidelity, we adopt CSD (Somepalli et al., 2024) to measure the style similarity between videos generated by the target model and those generated by the source model with same LoRAs. Please refer to Appendix C.2 for more details.

### 5.2. Main Results

Table 1 reports the quantitative comparison between direct LoRA reuse and CASA-based transfer across different settings. Overall, CASA consistently improves or maintains generation quality while achieving higher style fidelity, as reflected by Quality Score and CSD. Notably, the gains in CSD are more pronounced across most settings, indicating that CASA more reliably preserves LoRA-induced style under cross-model transfer and better recovers the intended adaptation effect. At the same time, the improvement in Quality Scores suggests that CASA effectively mitigates

*Table 2.* Ablation study on restoration (R) and arbitration (A), the two core components of CASA. Results are reported in terms of Quality and CSD. Best results are highlighted in bold.

| R | A | Steamboat-Willie | | Jinx-v2 | |
|---|---|---|---|---|---|
| | | Quality | CSD | Quality | CSD |
| ✗ | ✗ | 1.27 | 78.35 | 1.46 | 68.17 |
| ✓ | ✗ | 1.23 | 80.98 | 1.45 | 70.15 |
| ✗ | ✓ | **1.60** | 77.68 | 1.49 | 67.56 |
| ✓ | ✓ | 1.58 | **81.49** | **1.51** | **70.28** |

*Table 3.* Comparison of direct LoRA reuse, ProLoRA and CASA on FastWan2.1-T2V-1.3B. Best results are bold.

| Method | Steamboat-Willie | | Jinx-v2 | |
|---|---|---|---|---|
| | Quality | CSD | Quality | CSD |
| Direct Reuse | 1.27 | 78.35 | 1.46 | 68.17 |
| ProLoRA | 1.30 | 60.15 | **1.63** | 52.30 |
| CASA | **1.58** | **81.49** | 1.51 | **70.28** |

structural collapse during generation, leading to more coherent and stable video outputs. This trend is consistent across different LoRAs, highlighting the robustness of the proposed method under varying adaptation patterns. We further observe that the magnitude of improvement on 14B models is generally smaller than that on 1.3B models, indicating that larger models may inherently possess a more stable generative space and are therefore less sensitive to routing interference, and thus suffer less structural degradation introduced by such conflicts. This observation is also consistent with our analysis that stronger base models tend to better absorb perturbations, reducing the relative impact of transfer-induced interference.

## 5.3. Ablation Study

We conduct ablation studies on FastWan2.1-T2V-1.3B to evaluate the necessity and individual roles of the two core components of CASA, restoration (R) and arbitration (A). As shown in Table 2, using either component alone leads to suboptimal behavior. Restoration without arbitration improves style similarity but may degrade generation stability, as it introduces LoRA signals into sensitive routing pathways without accounting for potential conflicts. Conversely, arbitration without restoration preserves stability at the cost of weaker LoRA recovery, since it primarily constrains dominant pathways but does not sufficiently reinstate LoRA functionality in under-activated regions. Combining both components achieves the best overall performance across different LoRAs, consistently yielding higher CSD while maintaining strong quality. These results support our analysis that effective LoRA transfer on distilled video diffusion models requires both functional restoration in non-dominant regions and interference-aware arbitration in dominant routing pathways.

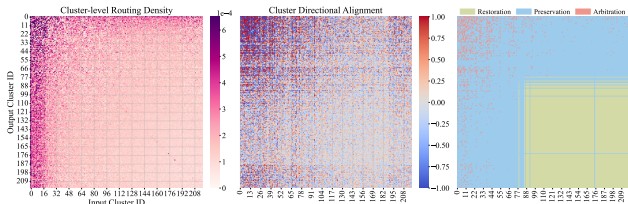

*Figure 7.* Visualization of routing energy density induced by FFT, directional alignment of FFT and LoRA, and different regions intervened by CASA for the same layer.

## 5.4. More Analysis

**Comparison with ProLoRA.** ProLoRA (Farhadzadeh et al., 2025b) is a representative data-free LoRA transfer method originally developed for image generative models, which decomposes LoRA updates into the subspace and null space of the source model and projects them onto the corresponding subspaces of the target model. We compare CASA with ProLoRA on FastWan2.1-T2V-1.3B, and report the result in Table 3. As presented, ProLoRA exhibits a substantial drop in CSD across both LoRAs, suggesting a pronounced loss of LoRA functionality and reduced effectiveness of the transferred adaptation. In contrast, CASA consistently improves CSD while maintaining generation quality. The degradation observed with ProLoRA aligns with our analysis that dominant routing pathways shaped by full fine-tuning play a critical role in VDMs, and that effective LoRA reuse requires interference-aware arbitration rather than uniform projection, particularly in regions with concentrated routing interactions.

**Visualization of CASA Intervention Regions.** Figure 7 illustrates how CASA intervenes in different regions of the routing space. The left panel shows the cluster-level routing energy density induced by FFT, where a small number of clusters exhibit substantially higher energy, indicating concentrated generative pathways. The middle panel reports the cluster-level directional alignment between FFT-induced and LoRA-induced routing. We observe heterogeneous alignment patterns, ranging from strong agreement to strong opposition, with pronounced structure in regions associated with high routing density. The right panel overlays the regions where CASA applies restoration, preservation, and arbitration[5]. Restoration is primarily applied to low-density regions, where routing interactions are weak and LoRA-induced behavior can be safely recovered with minimal risk of interference. In contrast, arbitration is selectively triggered in high-density regions with strong directional alignment, where over-activation might harm the generative space and thus requires careful regulation.

---

[5]Since arbitration operates at pixel-level, we regard blocks with more than $50\%$ pixels within it detected with over-activation risk as arbitrated.

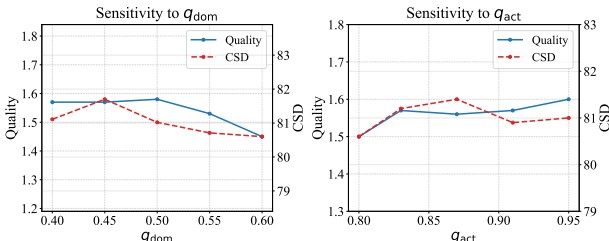

*Figure 8.* Sensitivity analysis of $q_{\text{dom}}$ and $q_{\text{act}}$.

**Sensitivity Analysis.** CASA introduces two quantile-based hyperparameters: $q_{\text{dom}}$ for identifying spectrally dominant routing regions, and $q_{\text{act}}$ for selecting routing entries requiring arbitration due to potential over-activation. To evaluate the robustness of CASA with respect to these thresholds, we conduct a sensitivity analysis by varying one parameter while fixing the other, and report both generation quality and style fidelity. Figure 8 shows that CASA maintains stable performance across a broad range of values. For $q_{\text{dom}}$, moderate values (around 0.45–0.50) achieve a good balance between generation quality and CSD. Smaller values lead to overly conservative intervention, limiting LoRA recovery, while larger values underestimate dominant pathways and allow residual interference. Similarly, CASA is robust to variations in $q_{\text{act}}$, as generation quality remains stable and CSD varies only slightly, indicating that the arbitration mechanism is not sensitive to precise threshold tuning. Overall, these results show that CASA does not require fine-grained hyperparameter tuning and remains effective across a wide range of threshold settings.

**Execution Time.** We report the execution time of CASA and its preprocessing steps across different model scales, with all experiments conducted on a single NVIDIA RTX 4090 GPU and excluding model and data loading overhead. CASA involves two one-time preprocessing stages and a lightweight per-LoRA transfer. First, singular value decomposition (SVD) of the source model's Transformer weights is computed once and reused across different LoRAs and target models, taking several tens of seconds for Wan2.1-T2V-1.3B and approximately 36 minutes for Wan2.1-T2V-14B. Second, we compute the FFT-induced weight drift $\mathbf{C}_{\text{fft}}$ by projecting the difference between target and source weights onto the source singular bases. This step also runs once per source–target pair and takes tens of seconds for the 1.3B model and about 12 minutes for the 14B model. Given these precomputed components, CASA performs LoRA transfer efficiently, requiring around 5 seconds per LoRA for the 1.3B model and about 1 minute for the 14B model. Overall, the computational cost is dominated by the one-time preprocessing, while the per-LoRA overhead remains minimal, making CASA practical for scenarios involving repeated LoRA reuse across shared source and distilled models.

*Table 4.* Evaluation of CASA on HunyuanVideo-1.5 family.

| LoRA | Target Model | Method | Quality Score | CSD (%) |
|---|---|---|---|---|
| Retro-Anime | HunyuanVideo-1.5-480P -T2V-CFG-Distill | Direct Reuse | 1.77 | 75.83 |
| | | CASA | **1.82** | **77.52** |

**Extension on other Model Family.** To verify whether our findings are general behaviors of video diffusion models, we extend our analysis and evaluation to a different video diffusion model family. Specifically, we use HunyuanVideo-1.5-480P-T2V (Wu et al., 2025) and HunyuanVideo-1.5-480P-T2V-CFG-Distill (Wu et al., 2025) as the base and distilled models, together with Retro-Anime[6] as the style LoRA. On this new model family, we conduct the same analysis as in Section 3 and observe consistent mechanisms. In particular, both the distilled and LoRA-adapted models exhibit spectral rigidity with extremely small changes in singular values, reinforcing the stability of the underlying spectral structure. We also observe structured subspace perturbations, with stable head, block-wise mixing in the middle spectrum, and diffuse behavior in the tail, mirroring the patterns identified in the Wan family. At the routing level, we again find that high-interaction regions concentrate in head clusters, and that the interaction between LoRA and FFT can be either strongly aligned or strongly opposed, without a global bias. Details of these analyses can be found in Appendix D.2. We further evaluate our method on the new model family using models and LoRA mentioned above. As presented in Table 4, CASA outperforms direct reuse in both quality and style fidelity, indicating the effectiveness of our method on this model family. Overall, these analyses and results demonstrate that our findings generalize beyond the Wan family and capture common structural behaviors in video diffusion models.

## 6. Conclusion

We studied why LoRAs trained on base video diffusion models often fail when reused on distilled variants. Through a weight-space analysis, we uncovered a pronounced spectral rigidity in VDMs and showed that both full fine-tuning and LoRA primarily introduce structured routing patterns at the cluster level, with incompatibility arising from conflicting routing interactions within spectrally functional subspaces. Motivated by these insights, we proposed Cluster-Aware Spectral Arbitration (CASA), a data-free transfer method that restores LoRA routing in non-dominant regions while arbitrating updates on dominant pathways to prevent over-activation. Experiments across multiple distillation paradigms and model scales demonstrate that CASA improves both generation stability and style fidelity, enabling effective LoRA reuse on distilled VDMs without additional training or user data.

---

[6]Please see Appendix C.1 for details.

## Impact Statement

This paper presents work whose goal is to advance the field of Machine Learning. There are many potential societal consequences of our work, none of which we feel must be specifically highlighted here.

## Acknowledgement

This work was partially supported by The Hong Kong University of Science and Technology (Guangzhou) Kunpeng&Ascend Center of Cultivation. This work was supported by Guangdong Provincial Key Lab of Integrated Communication, Sensing and Computation for Ubiquitous Internet of Things (No.2023B1212010007). This work was supported by National Major S&T Special Project on New Generation Artificial Intelligence (Nos. 2025ZD0123500), National Natural Science Fund of China (Nos. 62506155), Provincial Natural Science Fund of Jiangsu (Nos. BK20251985), and Suzhou Municipal Leading Talents Fund (Nos. ZXL2025320).

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

# A. Additional Examples of Structured Perturbations of Singular Subspaces

In this appendix, we provide additional evidence to support the structured singular subspace behavior analyzed in Section 3.2. Specifically, we visualize the cosine similarity matrices between left singular bases before and after adaptation for a broader set of layers and model scales.

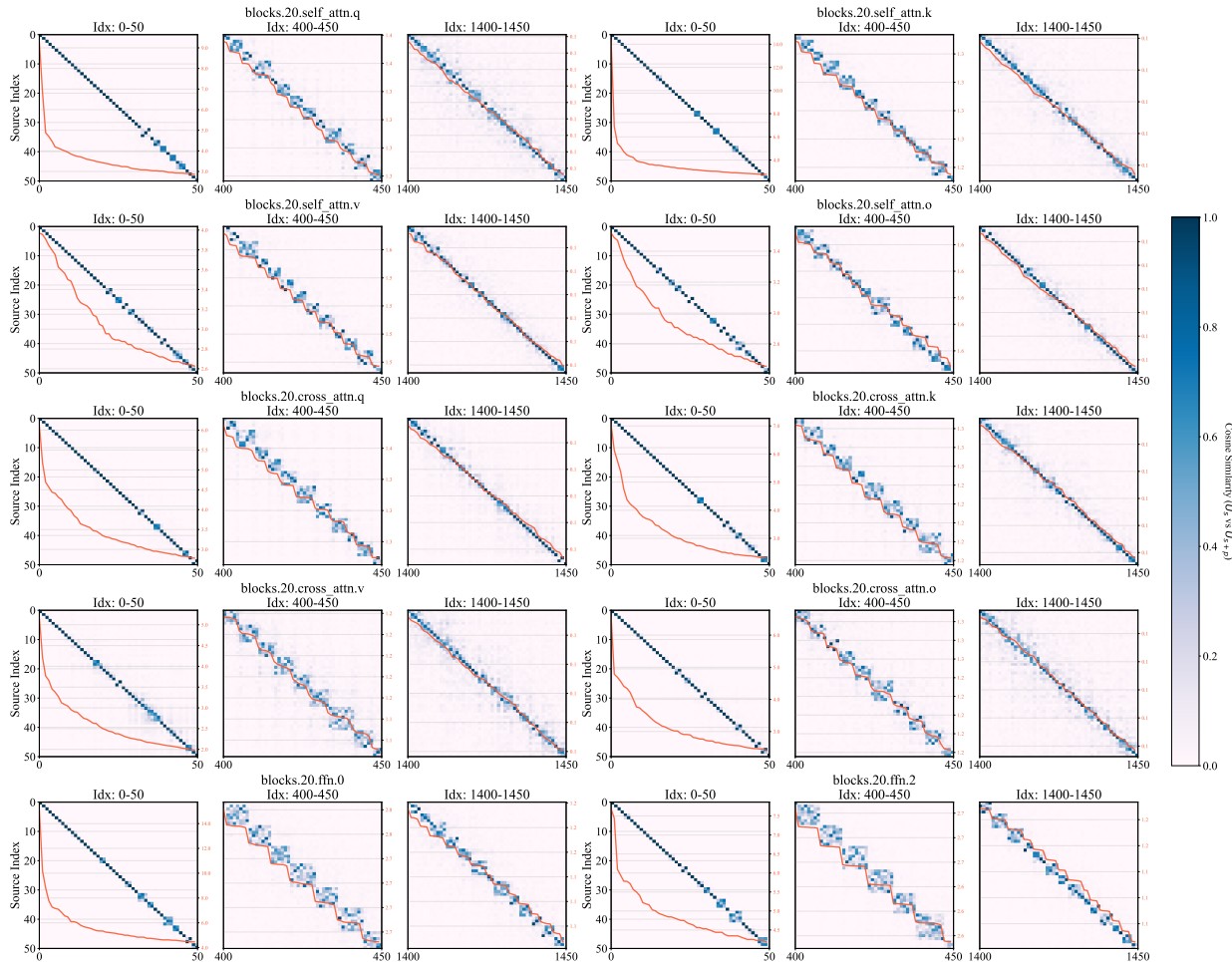

*Figure 9.* Similarity matrix of left singular bases before and after applying LoRA on transformer block 20 of Wan2.1-T2V-1.3B.

Figures 9 and 10 show representative results of layers from both Wan2.1-T2V-1.3B and Wan2.1-T2V-14B. For each projection matrix, we display the subspace similarity $|\mathbf{U}^\top \mathbf{U}'|$ over three spectral regions: the head of the spectrum, a middle region, and the tail. The curves indicate the corresponding singular value magnitudes, illustrating the relationship between spectral separation and subspace stability.

Across all examined layers and projection types, several consistent patterns emerge. First, singular directions associated with the largest singular values remain highly aligned after adaptation, resulting in sharply concentrated diagonal structures in the head region. Second, the middle spectrum consistently exhibits block-wise mixing patterns, where groups of neighboring singular vectors form coherent clusters with strong intra-cluster similarity. These blocks align closely with step-like plateaus in the singular value curves, indicating that local spectral gaps constrain subspace mixing. Third, the tail of the spectrum shows increasingly diffuse similarity patterns, reflecting near-degeneracy and weaker spectral separation in this regime. We observe that Wan2.1-T2V-14B shows less mixing in these regions, indicating a more robust generative space.

Importantly, these behaviors persist across self-attention (q/k/v/o), cross-attention, and feed-forward projections. This consistency suggests that the observed cluster-level organization is not an artifact of a specific layer or adaptation instance, but rather reflects a stable structural property of video diffusion models under fine-tuning.

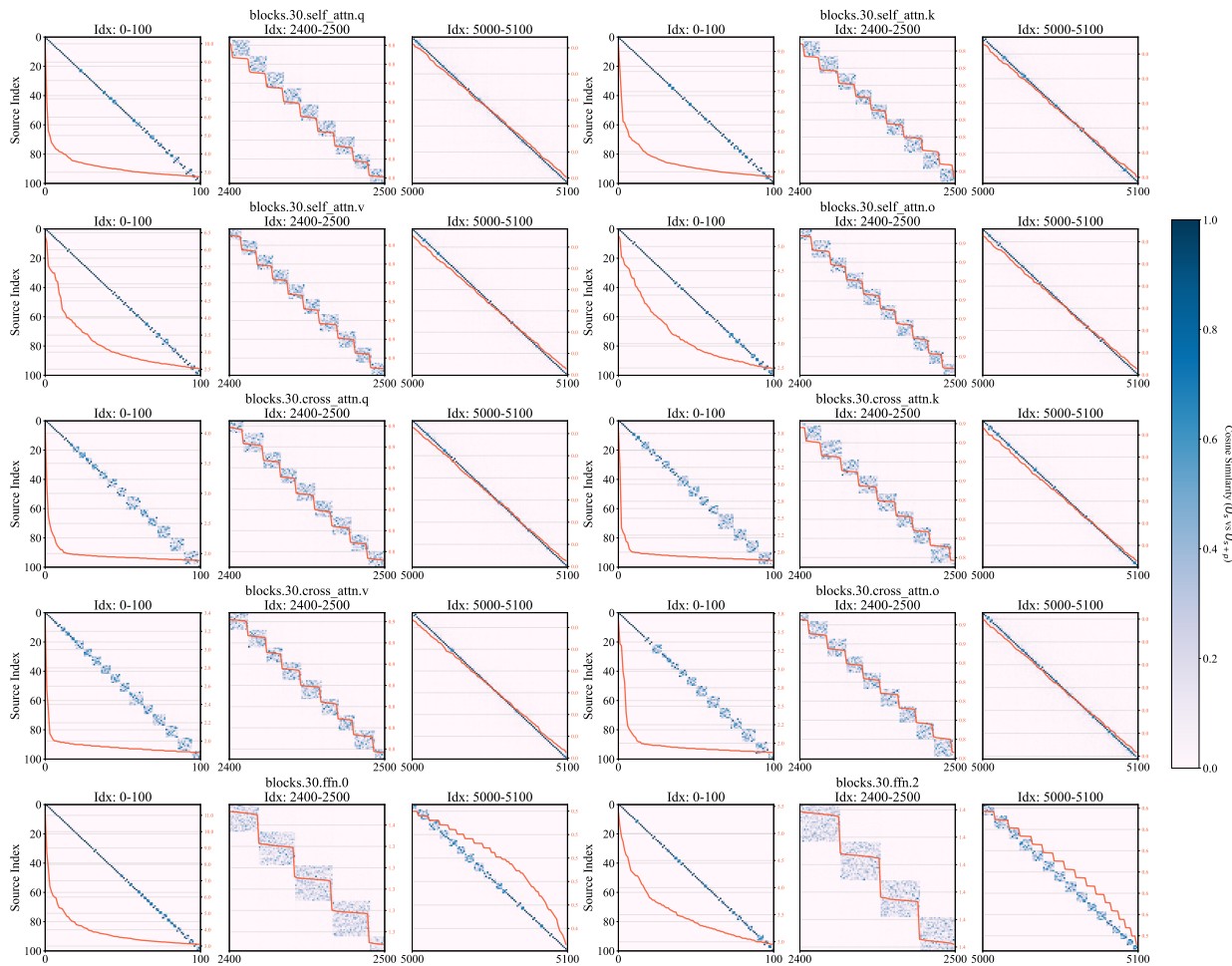

*Figure 10.* Similarity matrix of left singular bases before and after applying LoRA on transformer block 30 of Wan2.1-T2V-14B.

Together with the results in Section 3.2, these additional visualizations further support our claim that fine-tuning primarily induces structured perturbations of singular subspaces with strong cluster-level coherence, rather than arbitrary or unstructured rotations. This cluster-level organization provides a natural foundation for the routing-based analysis and arbitration strategy developed in the main paper.

## B. More analysis of Cluster-level Routing

### B.1. Cluster-level Routing Behaviors

In this appendix, we provide more visualization of the cluster-level routing behaviors introduced in Section 3.3. Specifically, we contrast the routing patterns induced by full fine-tuning and LoRA using both aggregated density profiles and cluster-to-cluster routing maps, covering self-attention, cross-attention, and feed-forward modules.

**Cluster-wise Routing Density Profiles.** Figure 11 illustrates the routing energy density associated with each singular cluster when acting as input (sender) or output (receiver). For each projection matrix, we report the root-mean-square (RMS) routing energy aggregated over all connections originating from or terminating at a given cluster.

Distributions of FFT and LoRA differ markedly in how routing energy is allocated. FFT displays a heterogeneous routing profile, where several clusters (typically in head regions) exhibit substantially higher routing energy relative to the rest. At the same time, many other clusters retain moderate decaying routing contributions, indicating that FFT does not collapse routing to a small subset of clusters.

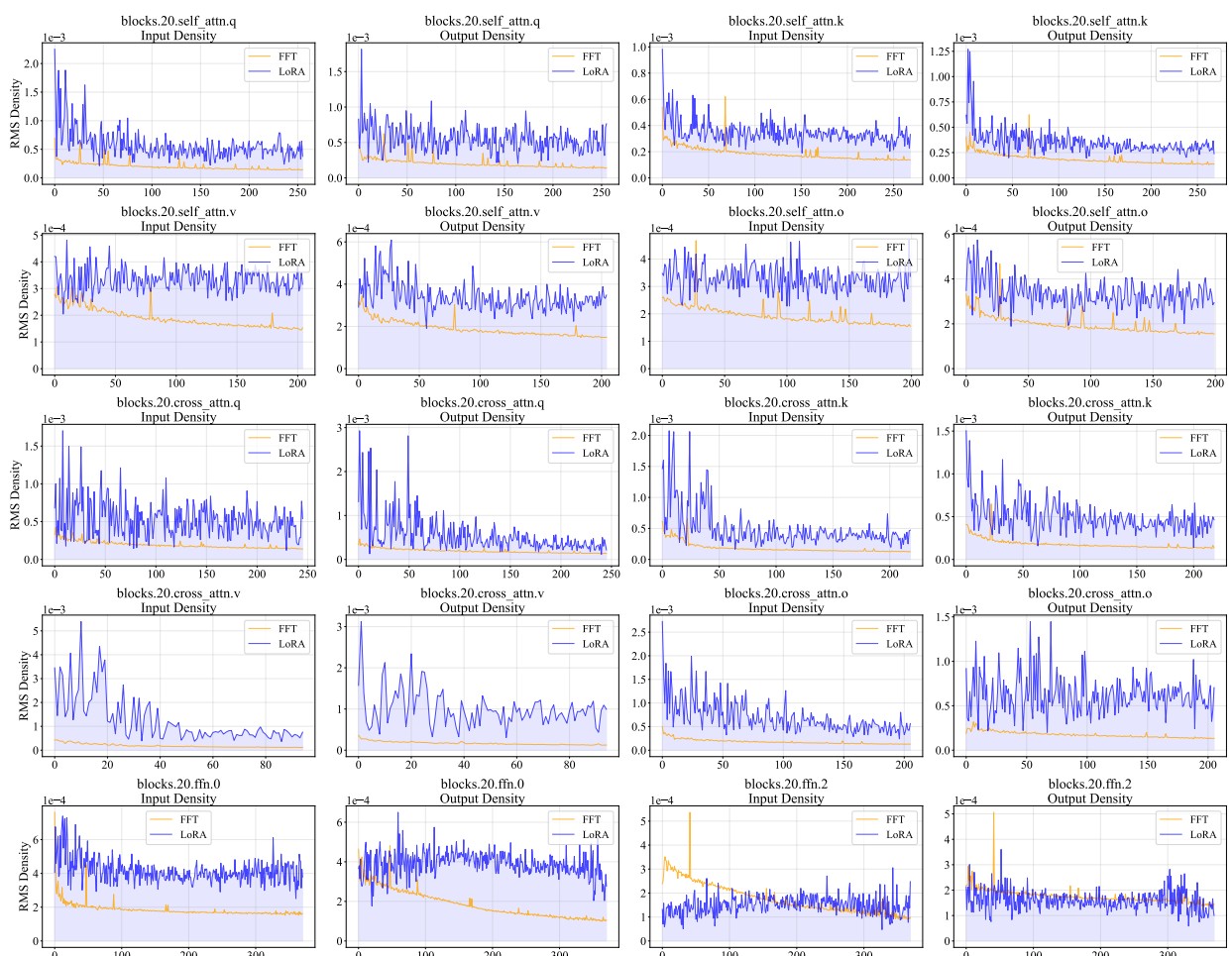

*Figure 11.* Cluster-level RMS energy density after adaptation with FFT and LoRA of all modules in block 20 of Wan2.1-T2V-1.3B.

By contrast, LoRA induces more uniformly distributed cluster-to-cluster interactions. Routing energy is spread across a larger number of cluster pairs, resulting in denser but less sharply peaked heatmaps. This distributed structure is consistent across attention and feed-forward modules.

**Cluster-to-Cluster Routing Structure.** Figure 12 further visualizes the routing behavior at the level of cluster pairs, where each entry represents the RMS routing energy between an input cluster and an output cluster. These heatmaps reveal the internal organization of routing beyond marginal cluster densities.

Under FFT, routing energy is typically highly structured and localized. Strong interactions concentrate in a limited set of cluster blocks, forming pronounced high-intensity regions that correspond to dominant generative pathways. Outside these regions, routing energy is uniformly weak, indicating that FFT selectively reinforces specific cluster-level connections.

By contrast, LoRA usually produces dense and widespread cluster-to-cluster interactions. Routing energy spreads broadly across the heatmap, with numerous cluster pairs exhibiting moderate interaction strength. This distributed pattern aligns with the view that LoRA operates by introducing cross-subspace couplings across a wide range of functional clusters, rather than amplifying a small number of existing pathways.

**Implications for Routing Interference.** Together, the density profiles and routing heatmaps provide complementary perspectives on the cluster-level routing behavior of FFT and LoRA. FFT establishes a centralized routing structure dominated by a small number of spectrally important clusters, while LoRA induces diffuse and globally distributed routing. When these two forms of routing coexist on the same model, interference tends to concentrate on the dominant cluster pathways emphasized by FFT, while remaining regions are weakly coupled.

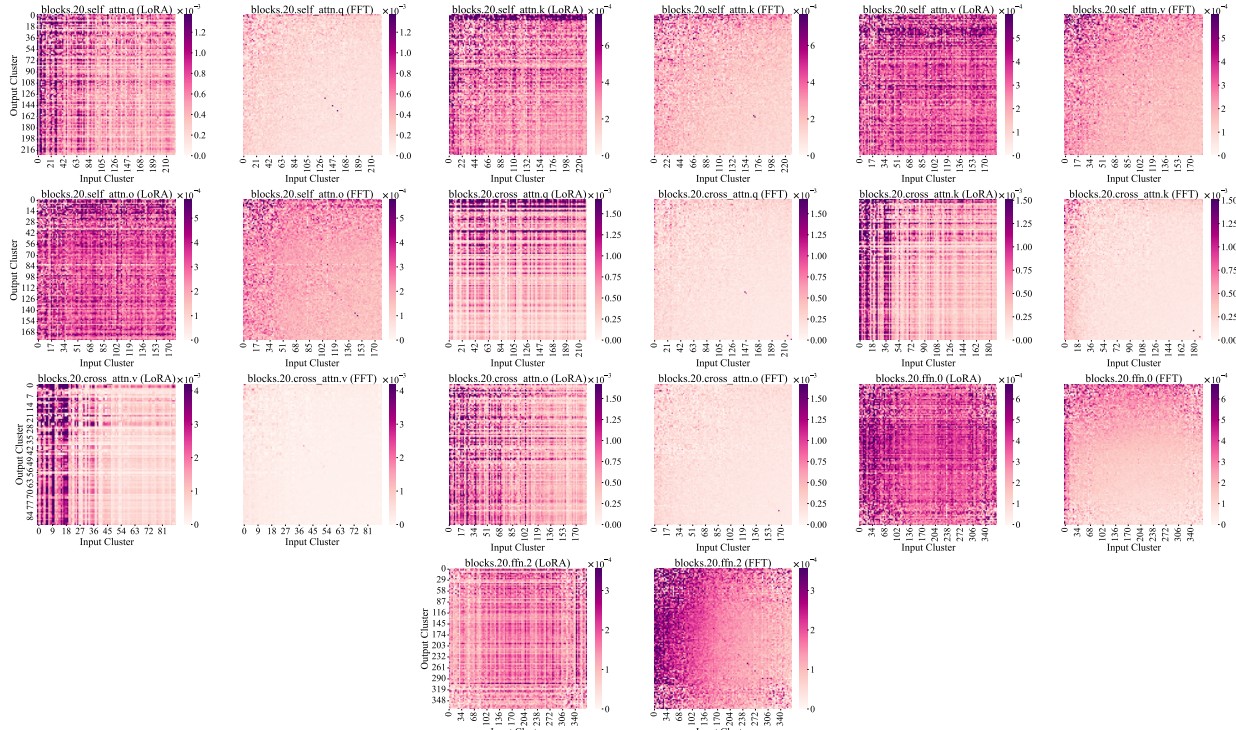

*Figure 12.* Cluster-level routing heatmap induced by FFT and LoRA of all modules in block 20 of Wan2.1-T2V-1.3B.

These observations provide empirical support for the cluster-level routing interference analyzed in Section 3.4, and motivate the need for selective, cluster-aware spectral arbitration when transferring LoRA to distilled video diffusion models.

## B.2. Generative Space from the Routing Perspective

### B.2.1. RELATIONSHIP BETWEEN DOMINANT CLUSTERS AND GENERATIVE SPACE

In Section 3.3, we show that the distilled-model drift $\Delta_{\text{fft}}$ induces highly structured cluster-level routing in the singular subspace. Here, we provide additional evidence that the high-energy routing clusters in $\mathbf{C}_{\text{fft}}$ are tightly connected to the model's effective generative space.

For a source weight matrix $\mathbf{W}_s = \mathbf{U}_s \mathbf{S}_s \mathbf{V}_s^\top$ and its distilled counterpart $\mathbf{W}_t$, we define the distilled drift

$$\Delta_{\text{fft}} \; = \; \mathbf{W}_t - \mathbf{W}_s, \tag{15}$$

and its routing representation in the source singular basis

$$\mathbf{C}_{\text{fft}} \; = \; \mathbf{U}_s^\top \Delta_{\text{fft}} \mathbf{V}_s. \tag{16}$$

Following Section 4.1, we partition the leading singular directions into clusters $\{\mathcal{G}_m\}_{m=1}^M$ (constructed on the top-$k$ subspace that captures $90\%$ spectral energy).

We quantify cluster-level routing density on the sender and receiver sides by

$$\rho_m^{\text{send}} \; = \; \frac{1}{|\mathcal{G}_m|} \sum_{j \in \mathcal{G}_m} \left\| \mathbf{C}_{\text{fft}}(:,j) \right\|_2, \qquad \rho_m^{\text{recv}} \; = \; \frac{1}{|\mathcal{G}_m|} \sum_{i \in \mathcal{G}_m} \left\| \mathbf{C}_{\text{fft}}(i,:) \right\|_2. \tag{17}$$

Given a quantile threshold $q \in [0,1]$, we define *dominant* sender/receiver cluster sets as

$$\mathcal{G}_{\text{dom}}^{\text{send}}(q) = \left\{ \mathcal{G}_m \mid \rho_m^{\text{send}} \geq Q_q\left( \{\rho_\ell^{\text{send}}\}_{\ell=1}^M \right) \right\}, \qquad \mathcal{G}_{\text{dom}}^{\text{recv}}(q) = \left\{ \mathcal{G}_m \mid \rho_m^{\text{recv}} \geq Q_q\left( \{\rho_\ell^{\text{recv}}\}_{\ell=1}^M \right) \right\}, \tag{18}$$

*Table 5.* Evaluation of videos generated by FastWan2.1-T2V-1.3B preserving cluster-level routing with energy density exceeding different quantiles. The column $q = 0$ indicates the videos are generated by the model originally.

| Metric | $q = 0$ | $q = 0.4$ | $q = 0.5$ | $q = 0.6$ | $q = 0.7$ | $q = 0.8$ | $q = 0.9$ |
|---|---|---|---|---|---|---|---|
| VideoAlign Quality Score | 2.30 | 2.26 | 2.25 | 1.93 | 1.87 | 1.84 | 1.69 |
| VBench Imaging Quality | 65.85 | 65.74 | 65.02 | 64.77 | 63.74 | 64.20 | 64.09 |

*Table 6.* Evaluation of videos generated by Rolling Forcing preserving cluster-level routing with energy density exceeding different quantiles. The column $q = 0$ indicates the videos are generated by the model originally.

| Metric | $q = 0$ | $q = 0.4$ | $q = 0.5$ | $q = 0.6$ | $q = 0.7$ | $q = 0.8$ | $q = 0.9$ |
|---|---|---|---|---|---|---|---|
| VideoAlign Quality Score | 3.08 | 3.14 | 3.12 | 3.05 | 2.91 | 2.76 | 2.63 |
| VBench Imaging Quality | 67.56 | 66.36 | 65.85 | 63.85 | 61.80 | 61.80 | 59.19 |

where $Q_q(\cdot)$ denotes the $q$-quantile of the given set. We then mark a routing entry $(i, j)$ as dominant if it involves a dominant sender or receiver cluster:

$$\mathcal{D}_q(i,j) = \mathbb{I}\big[\, i \in \mathcal{G}_{\text{dom}}^{\text{recv}}(q) \ \vee \ j \in \mathcal{G}_{\text{dom}}^{\text{send}}(q) \,\big]. \tag{19}$$

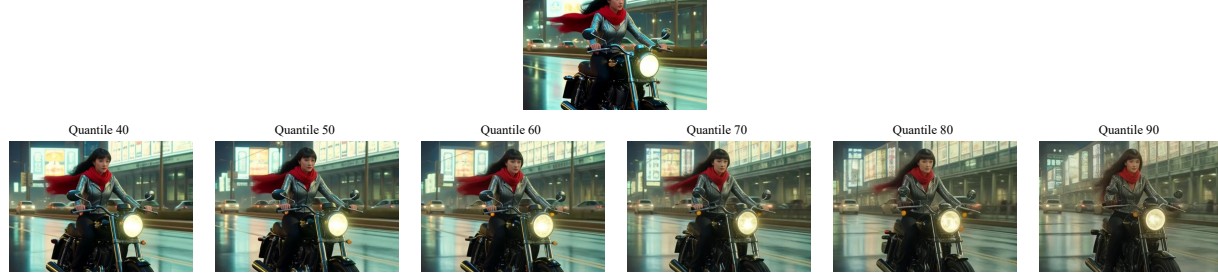

*Figure 13.* First frames extracted from videos generated by the distilled model, preserving only cluster-level routing with energy density exceeding different quantiles.

**Ablating Non-Dominant Routing and Constructing a Partial-Distilled Model.** To test how much of the distilled generative behavior is preserved by dominant routing clusters alone, we ablate the *non-dominant* part of $\mathbf{C}_{\text{fft}}$. Concretely, we form an ablation routing matrix

$$\mathbf{C}_{\text{abl}}^{(q)} = -\big(1 - \mathcal{D}_q\big) \odot \mathbf{C}_{\text{fft}}, \tag{20}$$

where $\odot$ denotes element-wise product. Projecting back to weight space gives the corresponding ablation update

$$\Delta_{\text{abl}}^{(q)} = \mathbf{U}_s \, \mathbf{C}_{\text{abl}}^{(q)} \, \mathbf{V}_s^\top = -\mathbf{U}_s\Big(\big(1 - \mathcal{D}_q\big) \odot \mathbf{C}_{\text{fft}}\Big)\mathbf{V}_s^\top. \tag{21}$$

We then obtain an ablated weight matrix by applying this update to the distilled model:

$$\mathbf{W}_t^{(q)} = \mathbf{W}_t + \Delta_{\text{abl}}^{(q)} = \mathbf{W}_t - \mathbf{U}_s\Big(\big(1 - \mathcal{D}_q\big) \odot \mathbf{C}_{\text{fft}}\Big)\mathbf{V}_s^\top. \tag{22}$$

By construction, $\mathbf{W}_t^{(q)}$ removes the routing contributions of non-dominant clusters, while retaining routing pathways associated with high-energy sender/receiver clusters. The special case $q = 0$ corresponds to the original distilled model.

**Results and Interpretation.** We evaluate videos generated by $\mathbf{W}_t^{(q)}$ using VideoAlign (Liu et al., 2025a) and VBench (Huang et al., 2024) Imaging Quality under different quantile thresholds. Table 5 and Table 6 report representative metrics, and Figure 13 visualizes sample frames. As $q$ increases, the generative quality gradually degrades, which is expected since more routing pathways are removed. However, a key qualitative observation is that the generated videos remain structurally coherent (e.g., subject layout and global motion stay organized) rather than collapsing into chaotic artifacts. This suggests that the dominant routing clusters in $\mathbf{C}_{\text{fft}}$ capture a substantial portion of the distilled model's effective generative space.

Notably, using a moderate quantile (around the median, i.e., preserving roughly half of the highest-density clusters) already maintains most of the perceived generative structure, even though fine details and sharpness may decrease. Together, these

findings support the connection between cluster-level routing density and the generative space, that high-energy routing clusters correspond to functionally critical pathways that anchor coherent generation, while the remaining lower-energy routing contributes more to refinement. A more detailed characterization of which generative functions map to specific dominant clusters is an interesting direction for future work.

### B.2.2. OVER-ACTIVATION OF DOMINANT ROUTING BLOCKS AND GENERATIVE FAILURE

The above results suggest that dominant routing clusters play a central role in anchoring the effective generative space of distilled VDMs. We further probe this connection by directly testing the effect of *over-activation* on these dominant routing pathways.

Specifically, we consider the cluster-level routing matrix $\mathbf{C}_{\text{fft}}$ and partition it into small sender–receiver blocks corresponding to cluster pairs $(\mathcal{G}_m, \mathcal{G}_n)$. For each block, we compute its routing energy density as the Frobenius norm normalized by block size. We then identify the top $5\%$ of the blocks with the highest routing energy density, which corresponds to the functional pathways that are the most utilized in the distilled model.

To simulate over-activation, we perturb these high-energy routing blocks by adding Gaussian noise sampled from $\mathcal{N}(\mu = 2, \sigma^2 = 5)$, while leaving all other routing entries as zero. The perturbed routing matrix is then projected back to the weight space using the source singular bases and added to the distilled model, yielding a modified model that differs from the original distilled model only through amplified activation along a small subset of dominant routing pathways.

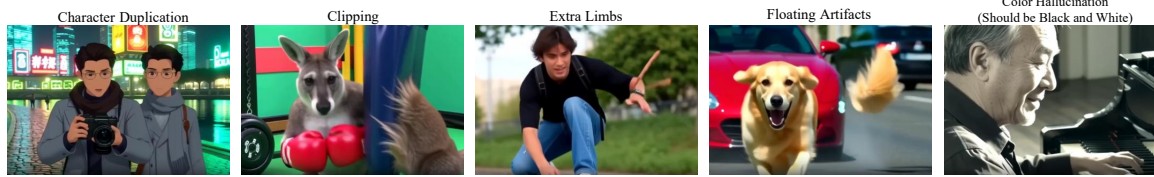

*Figure 14.* Examples of disrupted generation after over-activation in the cluster-wise interaction blocks exhibiting high energy density, induced by FastWan2.1-T2V-1.3B.

The effect of this targeted over-activation is immediate and severe. As illustrated in Figure 14, videos generated from the perturbed model exhibit pronounced structural failures, including but not limited to character duplication, limb hallucination, spatial clipping, and floating artifacts. Importantly, these failures arise despite the perturbation being confined to a very small fraction of routing blocks, highlighting the extreme sensitivity of the generative process to excessive activation along dominant pathways.

These observations provide direct causal evidence that dominant routing clusters are not only sufficient to sustain coherent generation (as shown in the previous subsection), but are also fragile to over-activation. Excessive amplification along these pathways destabilizes the generative space, leading to catastrophic structural artifacts. This behavior mirrors the failure modes observed when directly reusing LoRAs on distilled models, and strongly supports our interpretation of LoRA incompatibility as a form of spectral over-activation and interference within shared functional routing subspaces.

## C. Experimental Details

### C.1. LoRA Model Details

We provide the urls of the LoRAs we used in our experiments here.

- **Steamboat-Willie-1.3B**: https://huggingface.co/benjamin-paine/steamboat-willie-1.3b

- **Wan-LoRA-Arcane-Jinx-v2**: https://huggingface.co/Cseti/Wan-LoRA-Arcane-Jinx-v2

- **Film-Noir**: https://huggingface.co/Remade-AI/Film-Noir

- **Steamboat-Willie-14B**: https://huggingface.co/benjamin-paine/steamboat-willie-14b

- **Retro-Anime**: https://tensor.art/models/966330915603110253/HY1.5-Anime-Retro-Anime-Style-v1.0

## C.2. Details of Evaluation Metrics

**Generation Quality.**    We explain how we measure the quality of videos generated by directly reusing LoRA on the distilled VDM and by transferring it with CASA. We deploy VideoAlign (Liu et al., 2025a), a VLM-based reward model designed for video generation assessment. VideoAlign provides fine-grained reward signals by jointly evaluating Visual Quality, which measures frame-level image reasonableness and clarity, and Motion Quality, which evaluates dynamic stability, motion clarity, and the absence of artifacts such as flickering.

For each generated video, we compute both Visual Quality and Motion Quality scores using the pretrained VideoAlign reward model, and report their average as a unified video quality score. We then average the video-level scores across all evaluation samples to obtain the final quality score reported in our experiments. Compared to traditional handcrafted metrics, VideoAlign offers a learned, perceptually aligned evaluation that better captures common failure modes in video generation, making it particularly suitable for assessing generation stability under our setting.

**Style Fidelity.**    We explain how we measure the style similarity of videos generated by the source model and the target model with the same LoRA applied here. For each LoRA, we first generate 30 videos using a same group of prompts (generated by Gemini-3-Flash) using the source model and the target model. We next extract the first frame as the style proxy of the video, and use CSD (Somepalli et al., 2024) to generate the style embeddings of these frames. Finally, we compute cosine similarities for each pair of videos generated by the source model and target model using the same prompt, and obtain their average similarity as the final CSD-Score.

## C.3. Hardware and Environment

CASA can be implemented on Ascend 910B or 910C with PyTorch 2.6.0, SciPy 1.16.3, Safetensors 0.6.2 and NumPy 1.26.4.

# D. Additional Analyses

## D.1. Discussion and Guideline for Hyperparameter Selection

**top-$k$ Subspace Selection**    As analyzed in Section 3.2, the head and middle spectrum exhibit clear cluster-level organization, while the tail is near-degenerate and shows diffuse mixing. Including the tail would introduce many singular directions that cannot be reliably clustered, degrading routing analysis. In practice, selecting the top-$k$ subspace that captures 90% of the spectral energy works consistently across our settings. For other models, this threshold can be determined by inspecting the singular value spectrum and selecting the range where clear cluster-level structure (e.g., spectral plateaus with noticeable gaps) is present.

**Selection of $\tau$ for Clustering**    The threshold $\tau$ separates singular direction pairs based on their normalized interaction relative to spectral gaps. As shown in Section 3.2, intra-cluster pairs exhibit strong coupling with small spectral gaps, while inter-cluster pairs are weakly connected, leading to a natural separation in the interaction score. As a result, $\tau$ functions as a coarse structural separator rather than a sensitive hyperparameter. Empirically, we observe that clustering remains unchanged across a wide range of $\tau$ values (e.g., 1–10). In practice, a fixed value works across models without tuning.

**Quantile-based Thresholds**    CASA uses two quantile-based hyperparameters: $q_{\text{dom}}$ for identifying dominant routing regions and $q_{\text{act}}$ for detecting potential over-activation. For $q_{\text{dom}}$, we select it based on the relationship between routing energy and generation quality as analyzed in Appendix B.2.1. We observe that a subset of high-energy clusters is sufficient to maintain coherent generation. We therefore choose $q_{\text{dom}}$ as a quantile threshold that clusters with routing energy above this threshold preserve satisfactory generation quality. In practice, this threshold typically falls in the range of 0.45–0.55. For $q_{\text{act}}$, we base its selection on the observation that over-activation occurs in a small number of high-energy routing regions, as analyzed in Appendix B.2.2. We therefore use a high quantile to capture these sparse high-risk entries.

## D.2. Weight Space analysis on Hunyuanvideo-1.5

**Spectral Rigidity.**    As shown in Figure 15, on HunyuanVideo-1.5 family, we observe strong spectral rigidity showing as extremely small changes in singular values from both the distilled and LoRA-adapted models.

**Structured Perturbation.** As presented in Figure 16, we observe structured perturbation with stable head, block-wise mixing in the middle spectrum, and diffuse behavior in the tail on HunyuanVideo-1.5 family, mirroring the pattern we find on Wan family.

**Routing Interference.** As shown in Figure 17, at the routing level, we again observe that high-interaction regions concentrate in head clusters, and that the interaction between LoRA and FFT can be either strongly aligned or strongly opposed, without a global bias. We also observe some architecture-dependent differences. For example, in HunyuanVideo-1.5, routing interference in txt_attn layers is less concentrated than in img_attn, and img_mod/txt_mod exhibit much lower effective rank, consistent with their role as compact conditional scaling modules.

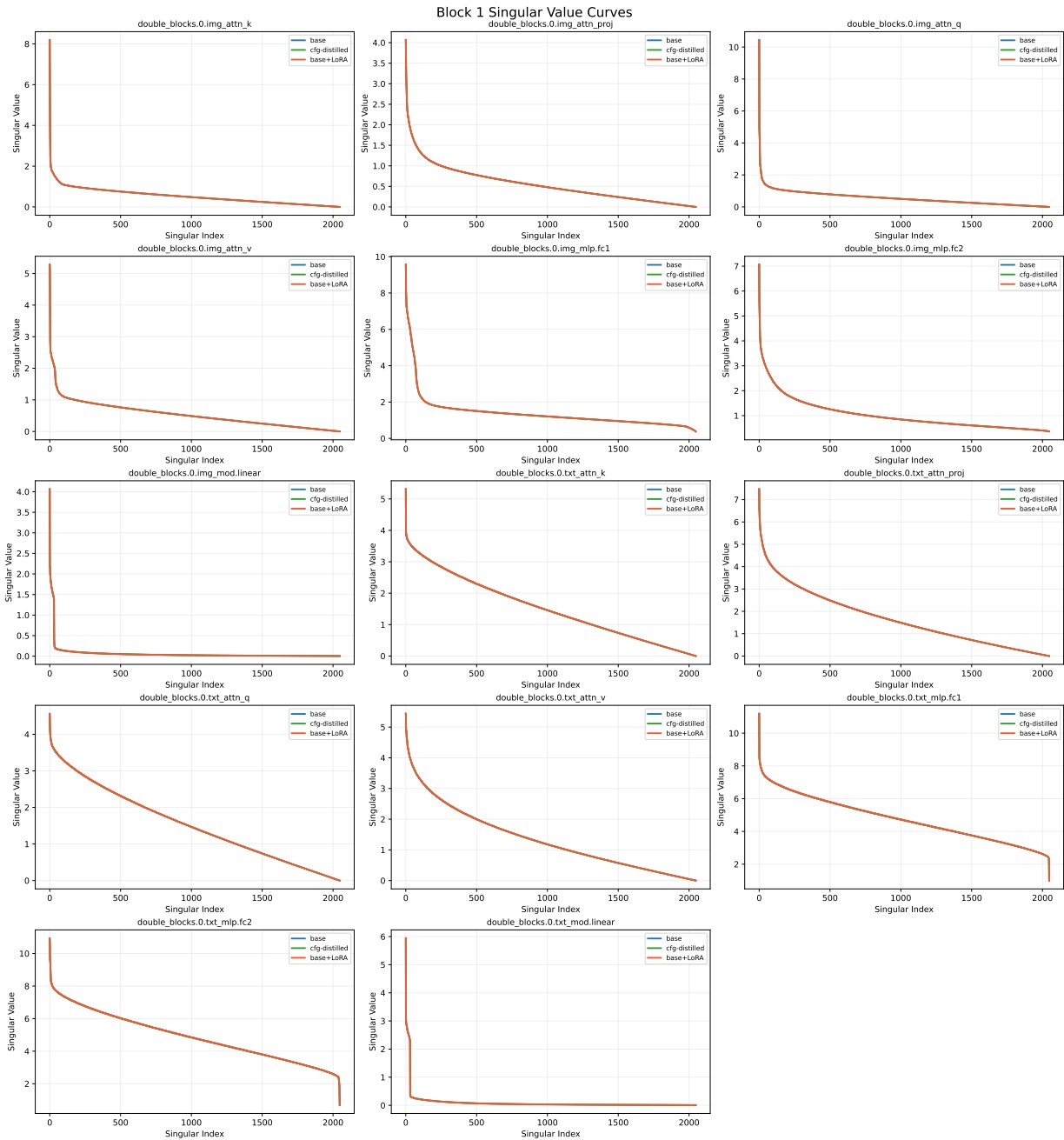

*Figure 15.* Spectral rigidity observed on HunyuanVideo-1.5 family.

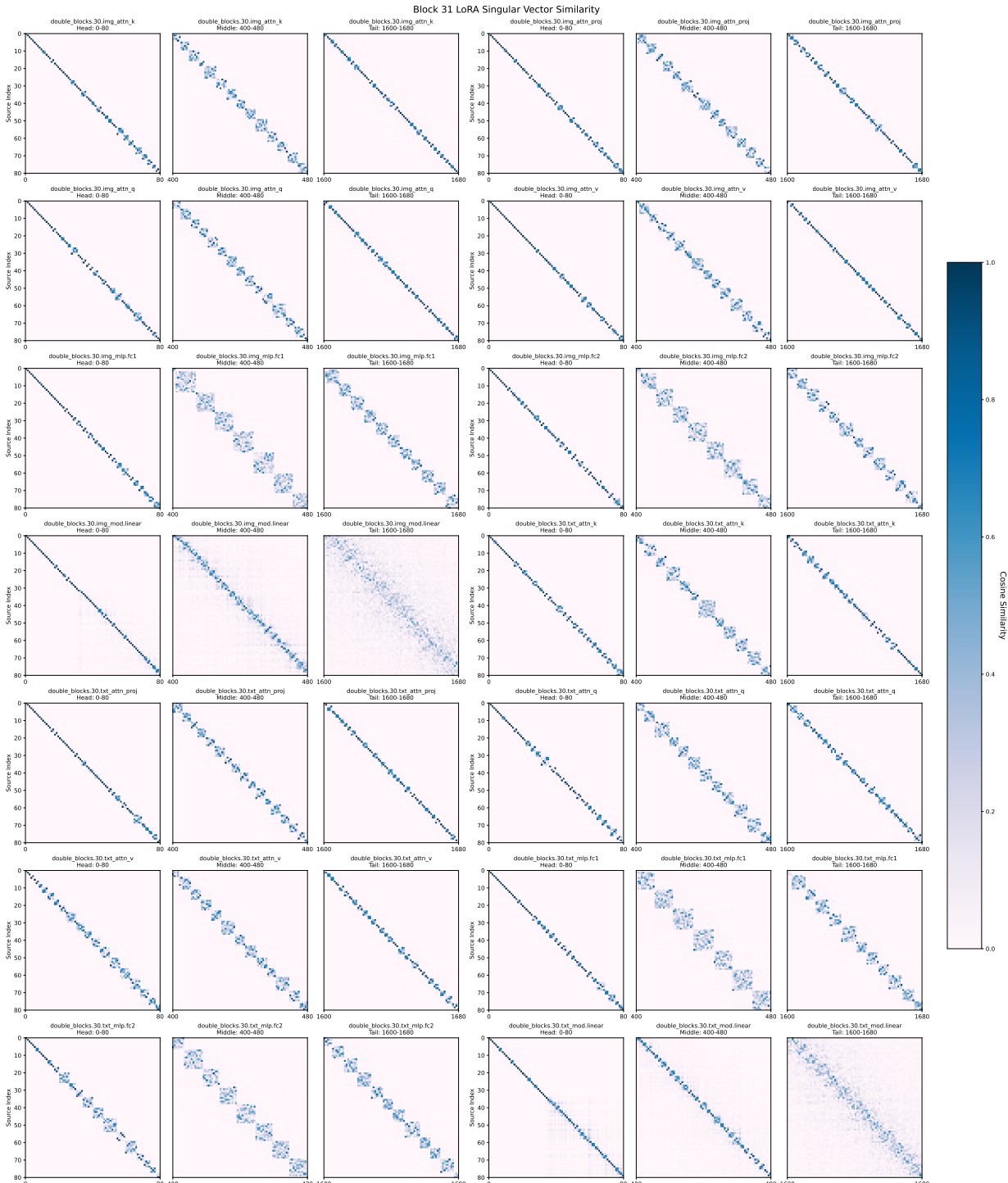

*Figure 16.* Structured perturbation observed on HunyuanVideo-1.5 family.

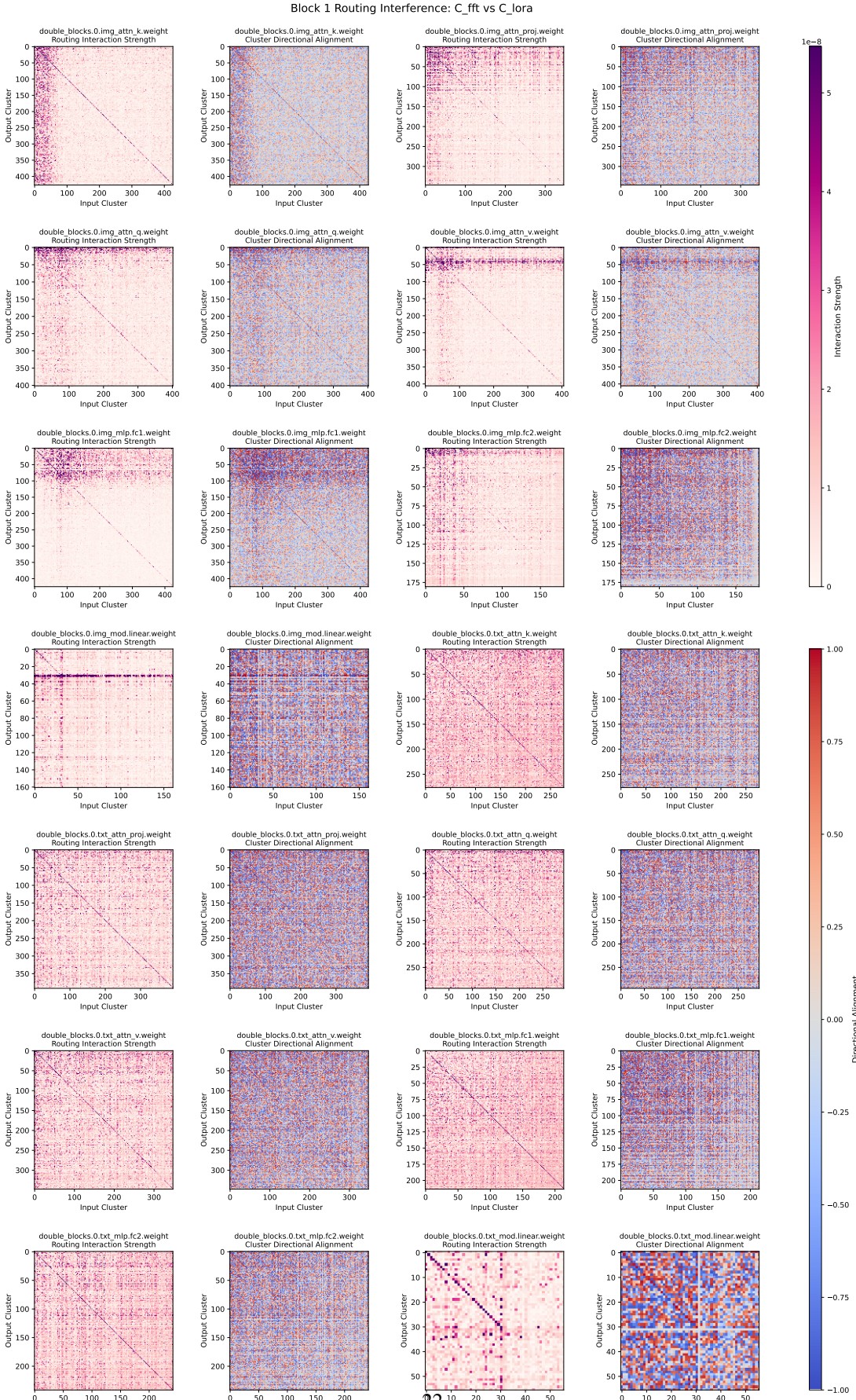

*Figure 17.* Routing interference observed on HunyuanVideo-1.5 family.