# OpenReview forum: "Exploring Data-Free LoRA Transferability for Video Diffusion Models"
_ICML.cc/2026/Conference — ICML 2026 regular_

### Official Review · Reviewer_HjPa · 2026-02-22

**Soundness:** 3
**Presentation:** 3
**Significance:** 3
**Originality:** 4
**Overall Recommendation:** 5
**Confidence:** 3

**Summary:**

The authors study the problem of reusing a LoRA adaptor from a base model to fine-tuned versions of the same model in Computer Vision, in particular in video generative models. The authors first identify that just naively reusing the LoRA adaptor does not work. Then, the authors investigate how this failure arises by investigating the effect of LoRA and fine-tuning to the spectrum of different layers of the neural network. The authors identify that the spectral decay is not affected by either finetuning or LoRA, however, the singular vectors corresponding to medium-to-low singular values are changed in different ways from LoRA and finetuning. The authors dive deeper into this phenomenon and identify a specific mechanism in which finetuning affects this space. Then, the authors use this insight to provide an algorithm for training-free adaptation of the original LoRA adaptor to the finetuned model. The algorithm achieves improved results over the baseline of applying the initial LoRA directly for video-diffusion.

**Compliance With Llm Reviewing Policy:**

Affirmed.

**Key Questions For Authors:**

- Please provide experiments with other video diffusion models (different base models).
- For your experiments above, if possible, experiment with different scales and see whether bigger models lead to worse results for your method, as you observed for the paper experiments.

**Limitations:**

N/A.

**Strengths And Weaknesses:**

Strengths:
- The authors study an interesting and important problem. It is true that many models have an initial common ancestor and hence being able to reuse LoRA adaptors from the original model to these derived models is an important ability that the community can benefit from.
- The authors do a very thorough and systematic analysis to identify why the naive application of the LoRA adaptor to the finetuned models fails. They identify interesting phenomena that are distinct to Vision (and not appear in the LLMs world) and they use clever techniques (the introduction of the C matrix) to understand this deeper and provide an interpretable analysis of the phenomenon.
- This analysis leads to **insights** about how to fix the problem; the authors use their findings to devise an algorithm that allows them to reuse an original LoRA adaptor to the finetuned models without any retraining and any data.
- The derived algorithms outperforms the baseline in the settings studied, showing that the method can actually be used in real settings and with real models.

Weaknesses:
- The derived algorithm has a lot of ad-hoc choices and tunable parameters. While certainly it is **a method** to improve the baseline, it is not clear to what extent this method is "optimal". The authors provide a (much-appreciated) sensitivity analysis in the Appendix; the algorithmic choices still seem a bit heuristic.
- The proposed method does not scale really well in bigger models.
- The writing can be a bit improved because in the abstract and the introduction a lot of terms are used without being introduced first (such as rigid spectrum) which makes it hard to parse what the paper is about without moving to the paper findinds.
- My biggest concern is the following; all experiments are performed using the Wan 2.1 video diffusion model. Hence, it is not clear whether this is a finding about LoRA and finetuning in the video domain, or a finding about how this particular model behaves. If it is the latter, the impact that the paper might have will be lower as the findings might not transfer to other (future) video models.

---

> ### Author Rebuttal · Authors · 2026-03-30
>
> Thanks for your time and helpful comments! We address each of your concerns below.
>
> ---
>
> > **Q1:** Experiments on Other Base Models
>
> Thank you for raising this important concern. We agree that it is critical to verify whether our findings are general behaviors of video diffusion models. To address this, we extend our analysis and evaluation to a different video diffusion model family. Specifically, we use HunyuanVideo-1.5-480P-T2V [1] and HunyuanVideo-1.5-480P-T2V-CFG-Distill [1] as the base and distilled models, together with a public community style LoRA. The findings are below.
>
> **Weight Space Analysis**
>
> On this new model family, we conduct the same analysis as in our paper and observe consistent mechanisms. In particular, both the distilled and LoRA-adapted models exhibit **spectral rigidity** with extremely small changes in singular values. We also observe **structured subspace perturbations**, with stable head, block-wise mixing in the middle spectrum, and diffuse behavior in the tail. At the routing level, we again observe that high-interaction regions concentrate in head clusters, and that the interaction between LoRA and FFT can be either strongly aligned or strongly opposed, without a global bias.
>
> We also observe some architecture-dependent differences. For example, in HunyuanVideo-1.5, routing interference in txt_attn layers is less concentrated than in img_attn, and img_mod/txt_mod exhibit much lower effective rank, consistent with their role as compact conditional scaling modules.
>
> Importantly, these differences mainly affect the distribution of routing across modules, rather than the underlying mechanism. The core phenomena we rely on, which are spectral rigidity and cluster-level routing interference, remain consistently observed across architectures.
>
> Additional figures illustrating these analyses are provided [here](https://anonymous.4open.science/r/CASA/Figures/).
>
> **Evaluation of CASA**
>
> We evaluate our method on the new model family using models and LoRA mentioned above. We present the results below.
>
> |   | Quality | CSD |
> |----------|---------|---------|
> | Direct Reuse | 1.77 | 75.83 |
> | CASA | **1.82** | **77.52** |
>
>
> These analyses and results demonstrate that our findings generalize beyond the Wan family. We will incorporate all of the above analyses and additional results in the next version.
>
>
> ---
>
> > **Q2:** Experiment with Different Scales and Degeneration on Larger Models
>
> Thank you for the suggestion. We agree that analyzing performance across different model scales is valuable.
> However, we are unable to perform a controlled cross-scale comparison within the HunyuanVideo-1.5 family, since it does not provide multiple parameter scales.
>
> That said, we can provide insights based on our observations on the Wan Family and the mechanism underlying CASA. Empirically, we observe that larger models tend to exhibit more stable generative behavior after distillation, with fewer visible artifacts under direct LoRA reuse. This is consistent with the intuition that larger models have higher capacity, leading to a **more robust generative manifold and reduced sensitivity to routing perturbations**.
>
> From the perspective of our analysis, CASA primarily addresses over-activation in high-interaction routing regions. When the base model is more stable, the severity of such over-activation effects is reduced, and therefore the relative improvement brought by CASA becomes smaller.
>
> At the same time, we observe that scaling the LoRA weights can amplify routing interference even in larger models, reintroducing noticeable artifacts. In such cases, CASA remains effective and leads to more significant improvements. To validate this, we conduct a small experiment by scaling the strength of the Film-Noir LoRA (1× vs. 3×) and evaluating generation quality on FastWan2.1-T2V-14B.
>
> *Table 1: Generation quality with different LoRA scales.*
> |   | scale=1 | scale=3 |
> |----------|---------|---------|
> | Direct Reuse | 1.90 | 1.74 |
> | CASA | 2.03 | 1.96 |
>
> We observe that increasing the LoRA scale leads to clear degradation under direct reuse, indicating stronger routing interference. In contrast, CASA consistently mitigates this effect and maintains higher generation quality.
>
> Overall, these observations indicate that while the magnitude of improvement may vary with model scale, the **underlying mechanism targeted by CASA remains present**, and CASA continues to provide consistent benefits when interference becomes significant.
>
> We will include a discussion on the relationship between model scale and CASA’s improvement in the revised version.
>
> ---
>
> [1] HunyuanVideo 1.5 Technical Report. arXiv:2511.18870, 2025.

---

> > ### Author Rebuttal · Reviewer_HjPa · 2026-03-31
> >
> > I am raising my score to Accept. I still think that a broader evaluation on more models and scales would benefit the paper.

---

### Official Review · Reviewer_HGkd · 2026-03-03

**Soundness:** 3
**Presentation:** 3
**Significance:** 3
**Originality:** 3
**Overall Recommendation:** 4
**Confidence:** 2

**Summary:**

This work investigates the challenge of transferring LoRAs to distilled video diffusion models without additional training or user data, proposing a novel solution, Cluster-Aware Spectral Arbitration (CASA). By examining how spectral interference between LoRA and fine-tuned models leads to style degradation and structural collapse, the paper offers a comprehensive weight-space analysis and introduces CASA, which effectively mitigates these issues. Experimental results show that CASA improves both the generation quality and style fidelity compared to direct LoRA reuse.

**Compliance With Llm Reviewing Policy:**

Affirmed.

**Final Justification:**

My concerns have been addressed. I will still keep my score.

**Key Questions For Authors:**

1.While CASA shows strong performance in the experiments, can it maintain efficiency when applied to large-scale video generation models? Is there a computational bottleneck when applied to larger models or more complex video content? This needs further exploration.

2.Hyperparameter selection could significantly impact performance across different LoRAs and video generation models. The paper does not go into much detail on how to choose and adjust these hyperparameters—are there detailed guidelines for this?

**Limitations:**

yes

**Strengths And Weaknesses:**

Strengths：This paper introduces an innovative Cluster-Aware Spectral Arbitration (CASA) method to address the issue of LoRA transferability in video diffusion models (VDMs). Through in-depth weight-space analysis, CASA effectively resolves the LoRA transfer failure in fine-tuned (FFT) models, enhancing both generation quality and style fidelity without requiring additional training or user data. Experimental results demonstrate significant advantages of CASA across various video diffusion models and LoRA settings.

Weaknesses：However, despite its strong performance in experiments, the paper does not sufficiently address the computational complexity and scalability of CASA in large-scale video generation models. Furthermore, the sensitivity to hyperparameters are not thoroughly discussed, which may limit its application in a broader range of scenarios.

---

> ### Author Rebuttal · Authors · 2026-03-30
>
> Thanks for your time and helpful comments! We address each of your concerns below.
>
> ---
>
> > **Q1:** Larger Model Scale and Computational Cost
>
> Thanks for pointing this out. We would like to clarify that our experiments already cover both 1.3B and 14B models. In particular, the 14B setting **already represents a genuinely large-scale open-source video diffusion model**, and therefore provides a meaningful test bed for evaluating the scalability of CASA.
>
> We note that the computational cost analysis is included in Appendix D.1. We apologize if this was not sufficiently visible in the main paper.
>
> To briefly summarize, CASA consists of two **one-time preprocessing** steps, including computing SVD for base and target models and obtaining $\mathbf{C}_\mathrm{fft}$, and one lightweight per-LoRA transfer step. On a single NVIDIA RTX 4090 GPU, for Wan2.1-T2V-1.3B, preprocessing takes on the order of tens of seconds, while the transfer step requires only ~5 seconds per LoRA. For Wan2.1-T2V-14B, preprocessing takes ~36 minutes (SVD) and ~12 minutes (drift), while each LoRA transfer takes only ~1 minute.
>
> Importantly, the dominant cost lies in the one-time preprocessing, which can be **reused across multiple LoRAs** for the same source–target model pair. Therefore, the amortized cost per LoRA remains low in practical reuse scenarios.
>
> As a result, **CASA does not introduce a per-adaptation computational bottleneck**. Its cost scales mainly with model size during preprocessing, but remains efficient for repeated LoRA transfer, which is the primary use case we target.
>
> ---
>
> > **Q2:** Discussion and Guideline for Hyperparameter Selection
>
> Thanks for raising this important question. We clarify the hyperparameter design in CASA and provide practical selection guidelines below.
>
> **top-$k$ Subspace Selection**
>
> As analyzed in Section 3.2, the head and middle spectrum exhibit clear cluster-level organization, while the tail is near-degenerate and shows diffuse mixing. Including the tail would introduce many singular directions that cannot be reliably clustered, degrading routing analysis.
>
> In practice, selecting the top-$k$ subspace that captures 90% of the spectral energy works consistently across our settings. For other models, this threshold can be determined by inspecting the singular value spectrum and selecting the range **where clear cluster-level structure (e.g., spectral plateaus with noticeable gaps) is present**.
>
> **Selection of $\tau$ for Clustering**
>
> The threshold $\tau$ separates singular direction pairs based on their normalized interaction relative to spectral gaps. As shown in Section 3.2, intra-cluster pairs exhibit strong coupling with small spectral gaps, while inter-cluster pairs are weakly connected, leading to a natural separation in the interaction score.
>
> As a result, $\tau$ functions as a coarse structural separator rather than a sensitive hyperparameter. Empirically, we observe that **clustering remains unchanged across a wide range of $\tau$ values (e.g., 1–10)**. In practice, a fixed value works across models without tuning.
>
> **Quantile-based Thresholds**
>
> CASA uses two quantile-based hyperparameters: $q_{\mathrm{dom}}$ for identifying dominant routing regions and $q_{\mathrm{act}}$ for detecting potential over-activation.
>
> For $q_{\mathrm{dom}}$, we select it based on the relationship between routing energy and generation quality as analyzed in Appendix B.2.1. We observe that **a subset of high-energy clusters is sufficient to maintain coherent generation**. We therefore choose $q_{\mathrm{dom}}$ as a quantile threshold that clusters with routing energy above this threshold preserve satisfactory generation quality. In practice, this threshold typically falls in the range of 0.45–0.55.
>
> For $q_{\mathrm{act}}$, we base its selection on the observation that **over-activation occurs in a small number of high-energy routing regions**, as analyzed in Appendix B.2.2. We therefore use a high quantile to capture these sparse high-risk entries.
>
> We further conduct sensitivity analysis under our setting, and the results are presented below (also shown in Appendix D.2):
>
> *Table 1: Sensitivity of CASA to different values of $q_{\mathrm{dom}}$.*
> |   | 0.40 | 0.45 | 0.50 | 0.55 | 0.60 |
> |----------|---------|---------|------------------|---------|---------|
> | Quality | 1.57 | 1.57 | 1.58 | 1.53 | 1.45 |
> | CSD | 81.1 | 81.7 | 81.0 | 80.7 | 80.5 |
>
> *Table 2: Sensitivity of CASA to different values of $q_{\mathrm{act}}$.*
> |   | 0.80 | 0.83 | 0.87 | 0.91 | 0.95 |
> |----------|---------|---------|------------------|---------|---------|
> | Quality | 1.50 | 1.57 | 1.56 | 1.57 | 1.60 |
> | CSD | 80.6 | 81.2 | 81.4 | 80.9 | 81.0 |
>
> The results indicate that CASA is generally robust to changes in $q_{\mathrm{dom}}$ and $q_{\mathrm{act}}$.
>
> We will incorporate these guidelines in the revised version.

---

> > ### Author Rebuttal · Reviewer_HGkd · 2026-04-01
> >
> > The authors' rebuttal has addressed my concerns.

---

### Official Review · Reviewer_p333 · 2026-03-12

**Soundness:** 3
**Presentation:** 3
**Significance:** 3
**Originality:** 3
**Overall Recommendation:** 4
**Confidence:** 4

**Summary:**

This paper studies the transferability of LoRAs trained on a base video diffusion model to its distilled variants without additional training or access to user data. The paper first analyzes the weight space of full fine-tuning and LoRA in video diffusion models, and argues that direct LoRA reuse fails because full fine-tuning and LoRA introduce incompatible routing patterns in singular subspaces, despite largely preserving the singular value spectrum. Based on this analysis, the paper proposes CASA, a data-free transfer method that restores LoRA routing in non-dominant regions while applying arbitration in dominant spectral regions to avoid over-activation. Experiments on Wan2.1-based 1.3B and 14B models, with both step-distilled and causal-distilled targets, show that CASA generally improves style fidelity and often improves generation quality over direct reuse and ProLoRA.

**Compliance With Llm Reviewing Policy:**

Affirmed.

**Key Questions For Authors:**

I hope the authors can clarify the issues in the weakness section (W1 and W2), as this would help strengthen the paper.

**Limitations:**

Yes

**Strengths And Weaknesses:**

Strengths

1.[Interesting Problem Setting]
The paper addresses a practically meaningful problem of reusing LoRAs on distilled video diffusion models without retraining or requiring user data, which is important because direct reuse often leads to severe artifacts and retraining for each target model is costly and impractical.

2.[Logical Analysis]
The proposed method is supported by a systematic weight-space analysis. The authors first study spectral rigidity, structured subspace perturbation, cluster-level routing, and routing interference, and then design CASA accordingly, which makes the paper technically coherent.

3.[Data-Free Transfer Setting]
The method operates in a training-free and data-free transfer setting, which is practically appealing in terms of both efficiency and privacy.

4.[Clear Presentation]
The paper is clearly written and well organized. The visualizations also help convey the intuition behind the proposed arbitration mechanism.

Weaknesses

1.[Some Design Choices Need Clearer Justification]
Some components in CASA depend on specific design choices, such as the top-k (90%) subspace selection, the thresholded perturbation graph for clustering, and the quantile-based thresholds for dominant regions and over-activation risk. The paper could better justify these choices and clarify their impact on performance.

2.[Experimental Scope Is Limited]
Although the paper evaluates both 1.3B and 14B models under step and causal distillation, the experiments are still mainly conducted on Wan2.1-based models with a limited number of public LoRAs. It remains unclear how well the conclusions generalize to other video diffusion families or a broader range of LoRA styles.

---

> ### Author Rebuttal · Authors · 2026-03-30
>
> Thanks for your time and helpful comments! We address each of your concerns below.
>
> ---
>
> > **Q1:**  Clearer Justification of Design Choices
>
> Thanks for pointing this out. We provide further justification and analysis for the key components in CASA below.
>
> **top-$k$ Subspace Selection**
>
> As analyzed in Section 3.2, the head and middle spectrum exhibit clear cluster-level organization, while the tail is near-degenerate and shows diffuse mixing.
> Including the tail region would introduce noisy and non-identifiable structures. Therefore, we restrict the region to the top-$k$ subspace to **avoid introducing a large number of singular directions that cannot be reliably clustered**, which would destabilize the clustering process and degrade the reliability of routing analysis.
>
> **Selection of $\tau$ for Clustering**
>
> The threshold $\tau$ separates pairs of singular directions based on their normalized interaction relative to spectral gaps. As shown in Section 3.2, intra-cluster pairs have strong coupling with small spectral gaps, while inter-cluster pairs are weakly connected. This leads to a natural separation in the interaction score, making $\tau$ a coarse separator rather than a sensitive parameter. Empirically, we observe that clustering remains unchanged across a wide range of $\tau$ (e.g., 1–10), indicating strong robustness.
>
> **Quantile-based Thresholds for Dominant Region and Over-Activation Identification**
>
> CASA introduces two quantile-based hyperparameters: $q_{\mathrm{dom}}$ for identifying spectrally dominant routing regions, and $q_{\mathrm{act}}$ for selecting routing entries with potential over-activation.
> We conduct sensitivity analysis on these thresholds, and present the results below (they are also reported in Appendix D.2).
>
>
> *Table 1: Sensitivity of CASA to different values of $q_{\mathrm{dom}}$.*
> |   | 0.40 | 0.45 | 0.50 | 0.55 | 0.60 |
> |----------|---------|---------|------------------|---------|---------|
> | Quality | 1.57 | 1.57 | 1.58 | 1.53 | 1.45 |
> | CSD | 81.1 | 81.7 | 81.0 | 80.7 | 80.5 |
>
> *Table 2: Sensitivity of CASA to different values of $q_{\mathrm{act}}$.*
> |   | 0.80 | 0.83 | 0.87 | 0.91 | 0.95 |
> |----------|---------|---------|------------------|---------|---------|
> | Quality | 1.50 | 1.57 | 1.56 | 1.57 | 1.60 |
> | CSD | 80.6 | 81.2 | 81.4 | 80.9 | 81.0 |
>
> The results indicate that CASA is generally robust to changes in $q_{\mathrm{dom}}$ and $q_{\mathrm{act}}$.
>
> ---
>
> > **Q2:** Generalization to Other Video Diffusion Model Families and Broader LoRA styles
>
> Thank you for raising this important concern. To broaden the scope of our work, we have conducted analysis and evaluation of CASA on a new video diffusion model family. Specifically, we use HunyuanVideo-1.5-480P-T2V [1] and HunyuanVideo-1.5-480P-T2V-CFG-Distill [1] as the base and distilled models, respectively. We use one public community style LoRA in this experiment. The findings are below.
>
> **Weight Space Analysis**
>
> On the new model family, we conduct the same analysis as in our paper. We find that both the distilled and LoRA-adapted models exhibit **spectral rigidity** with negligible singular value changes. We also observe **structured subspace perturbations**, with stable head, block-wise mixing in the middle spectrum, and diffuse behavior in the tail.
> At the routing level, we find that high-interaction regions are primarily concentrated in head clusters, and the interactions between LoRA and FFT can be either strongly aligned or strongly opposed without a global bias.
> These observations match the key mechanisms identified in our paper.
>
> However, there are also differences due to the model structure of HunyuanVideo-1.5.
> In particular, the txt_attn layers do not exhibit routing interference that is
> confined to the leading singular components as clearly as in img_attn layers.
> We also observe that img_mod and txt_mod have much lower effective rank and behave differently from attention/MLP layers, consistent with their role as compact conditional scaling modules.
>
> These differences do not affect our conclusions, as spectral rigidity and cluster-level routing interference, the core phenomena we rely on, remain consistently observed across architectures.
>
> Additional figures illustrating these analyses are provided [here](https://anonymous.4open.science/r/CASA/Figures/).
>
>
> **Evaluation of CASA**
>
> We evaluate our method on the new model family using models and LoRA mentioned above. We present the results below.
>
> |   | Quality | CSD |
> |----------|---------|---------|
> | Direct Reuse | 1.77 | 75.83 |
> | CASA | **1.82** | **77.52** |
>
>
> These analyses and results demonstrate that our findings generalize beyond the Wan family.
>
> ---
>
> We will incorporate all of the above justifications, analyses and additional results in the next version.
>
> [1] HunyuanVideo 1.5 Technical Report. arXiv:2511.18870, 2025.

---

> > ### Author Rebuttal · Reviewer_p333 · 2026-04-03
> >
> > Thanks for your rebuttal. I will keep my positive score.

---

### Official Review · Reviewer_iGBG · 2026-03-13

**Soundness:** 3
**Presentation:** 4
**Significance:** 3
**Originality:** 4
**Overall Recommendation:** 4
**Confidence:** 3

**Summary:**

When directly reusing LoRAs trained on a base video diffusion model to its distilled variants, it often leads to artifacts and style degradation due to weight-space mismatches between the base and distilled models. However, the underlying mechanisms of this phenomenon remain poorly understood. To fill the gap, this paper delves into the weight space and finds that both full fine-tuning (FFT) and LoRA preserve the singular value spectrurm but modify the singular subspaces through structured cluster-level routing patterns, which leads to spectral interference within shared functional clusters defined over singular subspaces when LoRAs are directly reused. To address this issue, the paper proposes Cluster-Aware Spectral Arbitration (CASA), a data-free method framework that restores LoRA routing in non-dominant spectral regions while arbitrating updates in dominant regions to avoid destructive interference.

**Compliance With Llm Reviewing Policy:**

Affirmed.

**Final Justification:**

I keep my positive score

**Key Questions For Authors:**

The same as the previous Weaknesses section

**Limitations:**

yes

**Strengths And Weaknesses:**

### Strengths
1. This paper investigates an important practical issue for LoRAs transfer from a base video diffusion model to its distilled variant and provides the first attempt to understand this phenomenon from the perspective of underlying mechanisms, which lays the foundation for subsequent practical.
2. The proposed CASA framework is practical and data-free, enabling LoRA reuse across distilled video diffusion models without additional training or access to user data.
3. The overall presentation is excellent.

### Weaknesses
1. As shown in Table 3, LoRA transfer baselines are not enough, there are more representative methods like: 1) LoRA-X: Bridging Foundation Models with Training-Free Cross-Model Adaptation (https://arxiv.org/abs/2501.16559) 2) Cross-LoRA: A Data-Free LoRA Transfer Framework across Heterogeneous LLMs (https://arxiv.org/abs/2508.05232) 3) Trans-LoRA: towards data-free Transferable Parameter Efficient Finetuning (https://arxiv.org/html/2405.17258v1).
2. The proposed method may introduce additional computational overhead, as it requires SVD and routing analysis for each layer in the weight space. However, the paper does not provide a clear experiment of the computational cost or scalability of CASA.

---

> ### Author Rebuttal · Authors · 2026-03-30
>
> Thanks for your time and helpful comments! We address each of your concerns below.
>
> ---
>
> > **Q1:** Insufficient Baselines
>
> Thanks for pointing this out. Following your suggestion, we conduct an additional experiment with **Cross-LoRA** [1], since it is also **data-free**, **training-free**, **post-hoc**, and performs transfer directly in the weight space, aligning with our setting.
>
> The results on the 1.3B model are shown below:
>
> *Table 1: Comparison of different transfer methods on FastWan2.1-T2V-1.3B. Best results are bold.*
>
> | LoRA | Method | Quality | CSD |
> | --- | --- | --- | --- |
> | Steamboat-Willie | Direct Reuse | 1.27 | 78.35 |
> | Steamboat-Willie | Cross-LoRA | 1.33 | 76.98 |
> | Steamboat-Willie | CASA | **1.58** | **81.49** |
> | Jinx-v2 | Direct Reuse | 1.46 | 68.17 |
> | Jinx-v2 | Cross-LoRA | 1.44 | 68.50 |
> | Jinx-v2 | CASA | **1.51** | **70.28** |
>
> From these results, we observe that CASA consistently outperforms the baselines. We further observe that Cross-LoRA exhibits limited effect on both metrics. We argue that this is because its foundational design **Subspace Alignment** is less effective in the video diffusion setting. Cross-LoRA applies Frobenius-optimal linear transformations to align the subspaces of the source and target models, and projects the LoRA updates into the aligned subspace. However, as shown in our analysis, in video diffusion models the source and target weights do not exhibit large global subspace shifts; instead, the main changes take the form of **structured perturbations**. As a result, explicit subspace alignment has limited room to help.
>
> For the other two methods, we believe they are less suitable for direct comparison in our setting:
>
> - **LoRA-X** [2] is designed to improve transferability during the LoRA
>   construction stage, rather than **post-hoc** transfer of an already trained community
>   LoRA, so it does not fully match our setting.
>
> - **Trans-LoRA** [3] requires synthesizing transfer data and then training on that data, so it is also not compatible with our **data-free, training-free** setting.
>
> We will clarify this baseline selection rationale explicitly and include the new Cross-LoRA results in the revised version.
>
> ---
>
> > **Q2:** Computational Cost of CASA
>
> Thank you for raising this issue. We would like to clarify that this analysis is already included in the current submission in Appendix D.1 (referenced in the main text as “Additional analyses on execution time and hyperparameter sensitivity are provided in Appendix D”). We apologize if this was not sufficiently visible.
>
> To briefly summarize the results here, CASA consists of two **one-time preprocessing** steps, including computing SVD for base and target models and obtaining $\mathbf{C}\_\mathrm{fft}$, and one **lightweight per-LoRA transfer** step. On a single NVIDIA RTX 4090 GPU, for Wan2.1-T2V-1.3B, the one-time SVD computation and FFT drift preprocessing each take on the order of tens of seconds, while the actual CASA transfer requires only ~5 seconds per LoRA. For Wan2.1-T2V-14B, the corresponding costs are ~36 minutes (SVD), ~12 minutes ($\mathbf{C}\_\mathrm{fft}$), and only ~1 minute per LoRA transfer.
>
> Importantly, the dominant cost lies in the one-time preprocessing, which can be **reused across multiple LoRAs** for the same source–target model pair. Therefore, the amortized cost per transferred LoRA remains low in practical reuse scenarios, which is a key design motivation of CASA.
>
> To improve clarity, we will explicitly highlight the computational cost analysis in the main text in the next version.
>
> ---
>
> [1] Cross-LoRA: A Data-Free LoRA Transfer Framework across Heterogeneous LLMs. arXiv:2508.05232, 2025.
> [2] LoRA-X: Bridging Foundation Models with Training-Free Cross-Model Adaptation. In Proc. of ICLR, 2025.
> [3] Trans-LoRA: towards data-free Transferable Parameter Efficient Finetuning. In Proc. of NeurIPS, 2024.

---

> > ### Author Rebuttal · Reviewer_iGBG · 2026-04-01
> >
> > Thanks for your clarification and additional experiment. I will keep my positive score.

---

### Decision · Program_Chairs · 2026-04-30

**Decision:**

Accept (regular)

**Comment:**

This paper studies an important and practical problem: reusing LoRAs trained on a base video diffusion model for its distilled variants without retraining or data access. The paper makes a meaningful contribution by not only identifying the failure mode of direct LoRA reuse, but also providing a coherent weight-space analysis that links this failure to structured spectral/subspace interference. Based on this analysis, the proposed CASA method offers a data-free and training-free transfer mechanism that improves both style fidelity and generation quality.

The reviewer consensus is clearly positive. Reviewers consistently appreciated the importance of the problem, the depth of the analysis, and the practical value of the proposed method. The rebuttal also addressed the main concerns effectively, including stronger baseline comparisons, computational cost, hyperparameter sensitivity, and evidence beyond the Wan family.